

Blue carbon stocks in Baltic Sea eelgrass (*Zostera marina*) meadows
Maria Emilia Röhr[1,2*], Christoffer Boström[1], Paula Canal- Vergés[3], Marianne Holmer[2]
[1]Åbo Akademi University, Faculty of Science and Engineering, Environmental and Marine
Biology, Artillerigatan 6, 20520 Åbo, Finland
[2]University of Southern Denmark, Department of Biology, Campusvej 55, 5230 Odense M,
Denmark
[3]Danish Shellfish Centre, DTU Aqua, Technical University of Denmark, Øroddevej 80, 7900
Nykøbing Mors, Denmark
* Corresponding author;
Emilia Röhr (mrohr@abo.fi)
Abstract
Although seagrasses cover only a minor fraction of the ocean seafloor, their carbon sink capacity
account for nearly one-fifth of the oceanic carbon burial and thus play a critical structural and
functional role in many coastal ecosystems. We sampled 10 eelgrass (*Zostera marina*) meadows
in Finland and 10 in Denmark to explore the seagrass carbon stocks (Corg stock) and the carbon
accumulation (Corg accumulation) in the Baltic Sea area. The study sites represent a gradient
from sheltered to exposed locations in both regions to reflect expected minimum and maximum
stocks and accumulation. The Corg stock integrated over the top 25 cm of the sediment averaged
627g C m$^{-2}$ in Finland, while in Denmark the average Corg stock was over six times higher (4324
g C m$^{-2}$). A conservative estimate of the total carbon pool in the regions ranged between 8.6-46.2
t ha$^{-1}$. Our results suggest that the Finnish eelgrass meadows are minor carbon sinks compared
to the Danish meadows, and that majority of the Corg produced in the Finnish meadows is
exported. Similarly, the estimates for Corg accumulation in eelgrass meadows in Finland (<



0.002- 0.033 t C y$^{-1}$) were over two orders of magnitude lower compared to Denmark (0.376-
3.636 Corg t y$^{-1}$). Our analysis further showed that > 40 % of the variation in the Corg stocks was
explained by sediment characteristics (density, porosity and silt content). In addition, the
DistLm analysis showed, that root: shoot- ratio of *Z. marina* explained > 12 % and contribution
of *Z. marina* detritus to the sediment surface Corg pool >10 % of the variation in the Corg stocks,
whereas annual eelgrass production explained additional 2.3 %. The mean monetary value for
the present carbon storage and sequestration capacity of eelgrass meadows at Finland and
Denmark, were 346 and 1862 € ha$^{-1}$, respectively. We conclude that in order to produce reliable
estimates on the magnitude of eelgrass Corg stocks, Corg accumulation and the monetary value
of these services, more Blue Carbon studies investigating the role of sediment biogeochemistry,
seascape structure, plant species architecture and hydrodynamic regime for seagrass carbon
storage capacity are in urgent need.
Keywords: Blue Carbon, eelgrass, seagrass,  carbon stock, carbon sequestration, carbon sink,
carbon burial
Introduction

The atmospheric inorganic carbon ($CO_2$) enters the ocean via gas-exchange processes at the
ocean-atmosphere interface. In the ocean dissolved inorganic carbon is fixed in photosynthesis
by primary producers, and released again through respiration. A large percentage of this fixed
carbon is stored and sequestered in the sediments of vegetated coastal ecosystems of which the
three globally most significant are saltmarshes, mangrove forests and seagrass meadows (Herr
et al. 2012). The carbon stored by these ecosystems is known as Blue Carbon (Duarte et al. 2005
Duarte et al. 2013a, Nellemann et al. 2009). Blue Carbon ecosystems function as carbon sinks, in
which the rate of carbon sequestered by the ecosystem exceeds the rate of carbon lost through
respiration and export.



Seagrass meadows play a critical structural and functional role in many coastal ecosystems (Orth et al. 2006). Although seagrass meadows only cover globally about 300 000 - 600 000 km² of the ocean sea floor, corresponding to 0.1 to 0.2 % of the total area, their carbon sink capacity may account for up to 18 % of the total oceanic carbon burial ( Gattuso et al.1998, Duarte et al. 2005, Kennedy et al. 2010, Fourqurean et al. 2012). A large portion of the carbon sequestered by seagrasses is stored in sediments with a conservative value of $10^{-10}$ t C in the top 1 meter of seagrass sediments (Fourqurean et al. 2012). Consequently, recent global estimates imply that seagrass sediments store almost 25 200- 84 000 t C km² (Fourqurean et al. 2012). More importantly, carbon in submerged sediments is stored for timescales of millennia while terrestrial soils are usually less stable and only sequester carbon up to decades (Hendriks et al. 2008).

The coasts of Scandinavia and the Baltic Sea are key distribution area for eelgrass (*Zostera marina* L.) meadows (Boström et al. 2002, Boström et al. 2014). The meadows extend from the fully saline (>30 psu) along the Norwegian coast to the brackish (5-6 psu) archipelago areas of Finland. This region is estimated to support > 6 000 individual meadows covering at least 1 500 – 2 000 km², which is four times more than the combined eelgrass area of the Western Europe (Spalding et al. 2003, Boström et al. 2014). Consequently, this region plays a key role in the coastal carbon dynamics, but we presently lack estimates of the role of eelgrass for carbon storage in temperate eelgrass sediments. To date, seagrasses are lost with accelerating rates (7 % $y^{-1}$) and it has been estimated that 29 % of global seagrass area has disappeared since the initial recording of seagrasses in 1879 (Waycott et al. 2009). This decline could have severe consequences on the total capacity of marine ecosystems to store and sequester carbon let alone the other ecosystem services seagrass meadows provide. As said, little is known about the magnitude of carbon emissions from degraded seagrasses ecosystems, not to mention its economic implications. Recent study (Pendleton et al. 2012) points out, that despite the



importance of these ecosystems to the global carbon budget, none of the three Blue Carbon
ecosystems have been included the global carbon market protocols.

Seagrasses exhibit marked differences in shoot architecture and grow under variable
environmental settings, making direct extrapolations between species and locations difficult.
Consequently, there is a pressing need to better understand which factors are causing variability
in carbon storage (Corg stocks) and accumulation (Corg accumulation) in seagrass sediments.
Indeed, recent studies show considerable influence of seagrass habitat setting, sediment
characteristics and species-specific traits on the variability in carbon storage capacity in
seagrass meadows (Duarte et al. 2013a, Lavery et al. 2013, Miyajima et al. 2015). Such
differences contribute to uncertainty in local and global estimates of the carbon storage capacity
and carbon dynamics in coastal seagrass areas.

In order to determine seagrass carbon stocks (carbon stored in living seagrass biomass and
sediments) and the capacity of seagrass meadows to sequester carbon, knowledge on the
sources of the accumulated carbon is also crucial. The different Corg sources vary in their
turnover rates compared to seagrasses (typically faster) and volumes of standing stock
(typically less) and thus affect the dynamics of the Corg stocks and accumulation. Seagrasses are
known to be isotopically heavier in $\delta^{13}$C compared to other potentially important sources of
Corg in the seagrass sediments, such as plankton, macroalgae, allochtonous carbon material,
seagrass epiphytes, and benthic microalgae (Kennedy et al. 2004, Kennedy et al. 2010, Fry and
Sherr 1984, Moncreiff and Sullivan 2001, Bouillon et al. 2002, Bouillon and Boschker 2006,
Macreadie et al. 2014). Thus, the stable isotope signals of seagrasses and other potential Corg
sources can be relatively easily and reliably used as a proxy for identification of the origin of
Corg in seagrass sediment carbon pool (Kennedy et al. 2010). When the sources of Corg in the
meadow have been determined, more reliable estimates on the capacity of the meadow to store
and sequester carbon can be made (Fry et al. 1977, Kennedy et al. 2004, Kennedy et al. 2010).





Unfortunately, the current knowledge base on how these factors interact and influence carbon
fluxes and storage is, at best, limited at both local and global scales.

In this study, we contrast storage, burial rates and sources of the accumulated carbon in eelgrass
meadows in two regions differing in salinity, temperature and seagrass productivity, namely
Finland and Denmark. Specifically we asked; (1) How large is the carbon storage capacity of
Baltic Sea eelgrass meadows? (2) Which are the environmental factors determining the
variability of carbon storage and accumulation at local and regional scales? (3) How do the
sediment characteristics influence the carbon storage of eelgrass meadows at local and regional
scales? (4) How much carbon is presently stored in Finnish and Danish eelgrass meadows
respectively, and what is the monetary carbon value of the historical eelgrass loss in Denmark?

Materials and methods

Study area

Plant and sediment samples were collected in June-September 2014 from 10 sites in Finland
(The Archipelago Sea) and 10 sites in Denmark (Funen and Limfjorden) (Fig. 1). The Baltic Sea
sediments are typically mineral sediments consisting of glaciofluvial deposits and only a small
fraction of the sediment carbon content consists of carbonates (Kristensen and Andersen 1987).
The inorganic carbon content in our samples was low (0.003 to 0.3 %DW, n= 10 per region) and
therefore carbonates were not removed from the sediment samples prior to the analysis. The
study sites in each region spanned a gradient from sheltered to exposed areas. The Archipelago
Sea of southwestern Finland is a shallow (mean depth 23 m), brackish (5-6 psu) coastal area
characterized by a complex mosaic of some 30.000 islands and skerries (Boström et al. 2006,
Downie et al. 2013). The region is heavily influenced by human pressures, especially



eutrophication, and exhibits naturally steep environmental gradients, as well as, strong
seasonality in temperature and productivity (Boström et al. 2014).

Limfjorden is a brackish water area in the Jutland peninsula connected to both North Sea and
Kattegat with salinity ranging from 17 to 35 psu. The Fjord has a surface area of ~1500 km² and
a mean depth of 4.7 m (Olesen and Sandjensen 1994, Wiles et al. 2006, Petersen et al. 2013).
Funen is located between the Belt Seas in the transition zone where waters from Baltic Sea and
Kattegat meet. The salinity of the area ranges between 10 and 25 psu and the annual mean water
temperature ranges from 10-15° C (Rask et al. 1999). This study was conducted in shallow (< 10
m) fjords around Funen. Also the Danish areas are heavily influenced by human pressures,
especially    eutrophication    from    intense    agricultural    farming.    (DMU;Danmarks
Miljøundersøgelser, 2003).

Field sampling

All samples were collected from depths of 2.5-3 m by scuba diving. At all sites, three replicate
sediment cores (corer: length: 50 cm, diameter: 50 mm) were taken randomly at a minimum
distance of 5 m from each other. The corer was manually forced to a depth of 30-40 cm. The
cores were capped in both ends under water, and kept in a vertical position during transport to
the laboratory. Eelgrass production and biomass were measured at all sites from four randomly
chosen locations within the eelgrass meadow. To insure statistical independence, each replicate
core was separated by distance of at least 15 m within the meadow. In the vicinity of each
sediment core, shoot density was counted using a 0.25 m² frame, and above- and belowground
biomass samples were collected with a corer (diameter 19.7 cm) and bagged underwater.
Sedimentation was measured at one exposed and one sheltered site in each region by deploying
two sediment traps with five replicate collection tubes (length: 115 mm, diameter: 28 mm).
Traps were positioned at a level corresponding to the upper canopy at a water depth of 2.5 m for



2 days. Additionally, when present, samples of plants and algae (drift algae, other angiosperms,
phytoplankton and epiphytes) considered most likely to be carbon sources in the eelgrass
meadows were collected from each site for identification and analysis of stable isotope
composition. Approximately ~ 10 g wet material was collected for each species. Annual eelgrass
production was determined from estimates of previous growth by applying the horizontal
rhizome elongation technique (Short and Duarte 2001). From each site, five replicate rhizome
samples with the longest possible intact rhizome carefully removed, were collected and
transported to the laboratory for further analysis.

Seagrass variables

In the laboratory, the above- and belowground biomass was separated and eelgrass leaves and
rhizomes were cleaned from epiphytes, detritus and fauna with freshwater and gently scrubbed
with a scalpel. All plant material was dried to constant weight (48 h in 60° C). The belowground
biomass was separated into living and dead rhizomes and dried separately. Only the living
rhizomes were used for the belowground biomass measurements while samples of both living
and dead rhizomes were used for analysis of POC and $\delta^{13}$C. The root: shoot-ratio was calculated
as the ratio between below- and aboveground biomasses of *Z. marina* samples. A pooled sample
of 2 youngest leaves from 10 randomly selected shoots were collected prior to drying from the
aboveground biomass samples and dried separately for analysis of particulate organic carbon
(POC) and stable isotopic composition of the organic carbon ($\delta^{13}$C). All samples were analyzed
by Thermo Scientific, delta V advantage, isotope ratio mass spectrometer. The measured isotope
ratios were represented using the $\delta$- notation with Vienna Peedee belemnite as reference
material.

Determination of annual eelgrass production was done by measuring length of each individual
internode of the rhizomes to the nearest millimeters. To obtain an estimate of the mean annual



production per site, internode length measurements of individual replicates (n= 5) were pooled.
Due to lack of two annual production peaks in both regions the annual production was estimated
based on the distance between shortest and longest measured internodes, assuming that they
represent the time point when the water temperature was at its minimum and maximum
average, respectively. The time points for the water temperatures were obtained from databases
of the Finnish and Danish Meteorological Institutes, respectively.

Sediment variables and sedimentation

In the laboratory, sediment samples were sliced into sections of 2-5 cm, where the upper 10 cm
layer was divided into 2 cm layers and the remaining part in 5 cm layers. From each subsample
visible plant parts and fauna were removed before the sediment was homogenized. From the 0-2
cm section a subsample of 20 ml was taken for grain size analysis by a Malvern Mastersizer 3000
particle size analyzer. The sediment silt content was calculated as the fraction with particle size
of 2-63 μm from the range of all particle sizes (Folk and Ward 1957). Sediment water content,
dry bulk density and porosity were determined from a subsample of 5 ml that was taken using a
cut-off 5 ml syringe and weighed before and after drying at 105°C for 6 h from all sediment
layers. The dried sediment samples were homogenized in a mortar and divided into two
subsamples from which one was used for analysis of sediment organic matter content (loss of
ignition: 4 h in 520°C), and the other for analysis of sediment $\delta^{13}$C and POC as described above
for the plant materials. Inorganic carbon content was low in sediments from both regions
(0.003-0.3 %DW) and considered insignificant compared to the organic fraction (1-2 order of
magnitude higher).

The material collected in the sediment traps was filtered on pre-weighed and combusted 50 mm
GF/C filters (Whatman) and dried at 60°C for 24 hours. Dried filters were weighed and each
filter was divided into two subsamples, one for analysis of organic matter content (LOI, 520°C, 4



h) and the other for $\delta^{13}C$ and POC as described above. Sedimentation rates were calculated
according to Gacia et al. (1999).

Corg stock and accumulation calculations

To estimate sediment Corg stock and Corg accumulation of Finnish and Danish eelgrass area we
used averages from 10 sites in each two regions in our calculations. The mean Corg stock
(obtained by depth integration of the POC mg C $cm^{-3}$ of 0-25 cm sediment layers) of the sampled
region was multiplied with estimated seagrass area of the region based on the most recent areal
estimates of seagrass distribution available in the literature (Boström et al. 2014) and given as g
Corg $m^{-2}$. In Denmark, the extrapolations are based on the minimum and maximum estimates of
the areal extent, respectively (673 and 1345 $km^2$, (Boström et al. 2014)). Results for carbon
burial (applied by multiplying the Corg stock, regional seagrass area and sedimentation rate
estimate from literature) in each area are given as Corg accumulation (t $y^{-1}$). Due to lack of long
term monitoring of sedimentation in eelgrass meadows, and seagrass meadows in general, we
used available minimum, average and maximum sedimentation rates in seagrass meadows
obtained from literature (Duarte et al. 2013b, Serrano et al. 2014, and Miyajima et al. 2015).

To calculate the total Corg pool in Danish and Finnish eelgrass sediments, we summed the
following three components: (1) the annual eelgrass carbon sequestration rate (1.66 t $ha^{-1}$:
Moksnes and Cole 2016), (2) the total POC (t $ha^{-1}$) in the average living aboveground and
belowground *Z. marina* tissue, and (3) the mean Corg stocks (t $ha^{-1}$) in eelgrass sediments in
Denmark and Finland, respectively. To calculate the present and lost economic value of eelgrass
carbon stocks, we used a value (40.3 € or 45 $) based on the social cost of carbon with 3 %
discount rate reported in United States Government Technical Support Document (2010) and
multiplied this value with the Corg stocks (tonnes). To estimate the Danish eelgrass losses over
the past 100 years in economic terms, we used the calculations above, but accounted for the





annually lost sequestration value by multiplying 1.66 t ha$^{-1}$ by 100 (166 t ha$^{-1}$). We used the most
recent loss estimates for Denmark for the period 1900-2000, assuming that the present
coverage constitutes 10% or 20% of the historical area (Boström et al. 2014).

Sediment carbon sources

The Isosource 1.3 isotope mixing model software (Phillips and Gregg 2003) was used to estimate
the contribution of different carbon sources to the sediment surface Corg pool. We ran the
Isosource model using the δ$^{13}$C values obtained from stable isotope analysis of *Z. marina* leaves,
living and dead rhizomes and for samples (n=1-5) of other abundant Corg sources within the
meadow (see above) with increments of 1 % and tolerance of 0.1. Numbers are given as
percentage contribution to the sediment surface carbon pool.

Data analysis

All statistical analyses were performed using the PRIMER 6 PERMANOVA+ package (Anderson
et al. 2008). A 2-factor mixed model was used, where sampling sites and region (FIN, DEN) were
used as fixed factors for the biological response variable (sediment organic carbon stock, g C m$^{-2}$
). Prior to analysis, the environmental predictor variables (degree of sorting, sediment dry
density, sediment water content, sediment porosity, sediment silt content, sediment organic
content, annual production, root: shoot-ratio, shoot density and percentage of *Z. marina* detritus
contribution to Corg were visually inspected for collinearity using Draftsman plots of residuals.
Due to autocorrelation between sediment variables (sediment porosity and dry density)
sediment water content was removed from the environmental variables. To achieve normality in
the retained environmental variables, data was log-transformed (log(X+1) and Euclidean
distance was used to calculate the resemblance matrix. The biological response variable (Corg
stock g m$^{-2}$) was square-root transformed and Bray-Curtis similarity was used to calculate the



abundance matrix. The relative importance of different environmental variables was determined
by use of DistLm, a distance-based linear model procedure (Legendre and Anderson 1999).
DistLm model was constructed using a step-wise procedure that allows addition and removal of
terms after each step of the model construction. AICc was chosen as information criterion as it
enables to fit the best explanatory environmental variables from of relatively small biological
dataset compared to number of environmental variables (Burnham and Anderson 2002). An
alpha level of significance of 95% (p<0.05) was used for all the analysis. All means are reported
as mean ± SE (SEM).

Results

Seagrass meadow and sediment characteristics

In general, the Finnish meadows were found on exposed sandy bottoms while the environmental
settings of the eelgrass meadows in Denmark were more variable (Fig. 2). Shoot density was
nearly equal in both regions, averaging at 417± 75 (shoots m$^{-2}$) in Finland and 418±32 (shoots
m$^{-2}$) in Denmark. In Finland variation between sites (112-773 shoots m$^{-2}$) was greater than in
Denmark (300-652 shoots m$^{-2}$). In Denmark the highest shoot density was found at the most
exposed site (Nyborg), while in Finland the highest shoot density was found at Sackholm, a fairly
sheltered site. The lowest shoot densities in Finland and Denmark were found in Tvärminne and
Løgstør, respectively. The mean aboveground biomasses were 101±3 and 145±5 (g DW m$^{-2}$) and
the mean belowground biomasses 79±5 and 148±13 g (DW m$^{-2}$) at Finnish and Danish sites,
respectively. In Denmark, the mean proportion of POC in above-ground and below-ground *Z.*
*marina* tissue was 35% and 29%, respectively, while the corresponding numbers for Finland
were 38% and 36%, respectively. Given an average total *Z. marina* biomass (above- and
belowground) of 293 (Denmark) and 180 g DW m$^{-2}$ (Finland), we estimate the Corg pool in
bound in living seagrass biomass to 0.94 and 0.66 t ha$^{-1}$ in Denmark and Finland, respectively.



Root: shoot-ratio was slightly lower in Finland (0.87±0.05) than in Denmark (1.14±0.12), and
varied between 0.29 to 3.29 and 0.15 to 6.45 in Finland and Denmark, respectively. The annual
production of eelgrass for Finland (average 524±62 g DW m$^{-2}$ y$^{-1}$) showed relatively low
variation between sites (270-803 g DW m$^{-2}$ y$^{-1}$) being lowest at Jänisholm and highest at
Ryssholmen. In Denmark, the mean annual eelgrass production rate was almost twice as high
(928±159 g DW m$^{-2}$ y$^{-1}$) with large variation (470-2172 g DW m$^{-2}$ y$^{-1}$). Production rates were
lowest and highest at Dalby and Visby, respectively (Table 1).

The sediment characteristics varied significantly between Finland and Denmark. There was a
significant difference ($F_{1, 9}$ = 14.7, p<0.003) between regions in terms of silt content, which was
generally lower at Finnish (6.3±1%) sites than at Danish sites (20.2±3.9%), although in Denmark
the variation between sites ranged from 0.8% at Nyborg to 31.6% at Thurøbund (Table 1, Fig. 2).
In Finland, the variation between sites was lower and ranged from 1.6% (Kolaviken) to 15.5%
(Sackholm). At the Finnish sites the mean sediment dry density was higher (1.30±0.04 g cm$^{-3}$)
compared to the Danish sites (1.1±0.1 g cm$^{-3}$), and the Finnish sites exhibited lower within-
region variability ranging from 1.1 g cm$^{-3}$ at Lyddaren to 1.5 cm$^{-3}$ at Långören, while the Danish
sites varied from 0.3 g cm$^{-3}$ at Thurøbund to 1.5 g cm$^{-3}$ at Visby. The Finnish sites showed
consistently lower pools of organic matter (LOI: 1.4±0.3% DW) compared to the average of
Danish sites (LOI: 3.9±1.5 % DW). Consequently, the mean water content was similarly lower in
Finland (20.9±0.4 %: range 16-29 %) than in Denmark (37.4±1.8 %: range 17-76 %) ranging
from 16.4 to 29.5 % in Finland to 17.2 to 76.0 % in Denmark (Table 1). Sediment porosity was
similar in both regions, and ranged from 0.25 to 0.3 in Finland, and from 0.20 to 0.40 in
Denmark. At the Finnish sites, the proxy that was used to estimate exposure (degree of sorting),
varied from 0.8 to 1.5 (φ), with Kolaviken being the most exposed and Ängsö being the most
sheltered site. In Denmark degree of sorting varied from 0.6 to 2.1 (φ,) with Nyborg and Visby
being the most exposed and sheltered sites, respectively.





Organic carbon stocks

The profiles of Corg stocks (g C cm$^{-3}$) in the upper 25 cm of the sediment showed marked differences both between and within the sampled regions. At the Finnish sites, where eelgrass typically grows at exposed locations, the sediment Corg stocks were low (<0.001-22.1 mg C cm$^{-3}$) and declined with depth at most of the 10 study sites (Fig. 3). At the Danish sites, however, the Corg stocks were more variable (<0.001 to 176.7 mg C cm$^{-3}$) both within and between the sites (Fig. 3). Corg stocks (g C m$^{-2}$, Fig. 4) were particularly high at one sheltered site in Funen, namely Thurøbund. This site is characterized by soft sediments with high organic content, high annual eelgrass production and high belowground biomass (Table 1). The lowest eelgrass Corg stocks in Denmark were found at two relatively exposed and sandy sites, namely Nyborg and Dalby (Fig. 4). The estimate of average total Corg stock in Finland ranged from 0.019±0.001 t (Table 2). Using minimum and maximum estimates of the eelgrass area in Denmark the estimates for mean total sediment Corg stock in Denmark were 2.937±0.005 or 5.868±0.014 t, respectively (Table 2).

The estimated total Corg stocks in the upper 25 cm of the sediments was on average 6.27 and 43.6 t ha$^{-1}$ in Finland and Denmark, respectively. Using an annual carbon sequestration value of 1.66 t ha$^{-1}$ (Moksnes and Cole 2016), the total pool of Corg in the *Z. marina* meadows (Corg bound in living biomass, sediment Corg stock and Corg sequestration) corresponds to 8.59 t ha$^{-1}$ (859 t km$^{-2}$) and 46.2 t ha$^{-1}$ (4620 t km$^{-2}$) and for Finland and Denmark, respectively. Using the social cost of carbon of 40.3 € t$^{-1}$ (United States Government 2010), the present economic value of eelgrass carbon in Finnish and Danish eelgrass meadows is estimated at 346 and 1862 € ha$^{-1}$, respectively. Using an average of these values (1104 € ha$^{-1}$) and the conservative estimate of the eelgrass acreage at the Baltic Sea (2100 km$^2$: Boström et al. 2014), we estimated the total monetary value of present carbon storage and sequestration capacity in the eelgrass meadows to be 231.9 millon €. Given the total eelgrass loss in Denmark for the time period 1900-2000 is





between 5381 (present area 20% of historical distribution) and 6053 km² (present area 10% of
historical distribution), this equals a Corg loss of 0.127 and 0.113 Gt, respectively. These areal
loss estimates corresponds to a lost economic value of 4565 to 5136 million €, for the minimum
and maximum areal estimates, respectively.

Corg accumulation

The estimates for annual Corg accumulation in the Finnish seagrass meadows were low (0.002,
0.016, 0.033 Mt y⁻¹), when applying sedimentation rates of 0.32, 2.02 and 4.20 mm y⁻¹,
respectively. Similarly, the sedimentation rates measured by use of sedimentation traps were
10–folds lower in Finland (3.6-7.7 gDW m⁻² d⁻¹) compared to Denmark (37.6-63.4 gDW m⁻² d⁻¹).
The low estimates of Corg accumulation in Finnish meadows were a result of low mean Corg
stocks and relatively small size of seagrass area in the region compared to Denmark (Table 2).
The estimates for Corg accumulation were generally higher for the Danish sites, but differed
between the two sub-regions Limfjorden and Funen. At the sampling sites around Funen, the
estimated corresponding Corg accumulation values were 0.139, 0.881 and 1.832 Mt y⁻¹, while in
Limfjorden the estimated Corg accumulation were lower (0.006, 0.038 and 0.079 Mt y⁻¹) and
similar to Corg accumulation estimates for Finland. Using upper and lower eelgrass areal
estimates, the estimates for total Corg accumulation (0.376, 2.373, 3.636 and 0.75, 4.741 and
9.859 Mt y⁻¹) in Denmark were more than four orders of magnitude higher than the estimated
total Corg accumulation in Finnish eelgrass meadows.

Carbon sources

The isotope mixing model showed that at all Finnish sites, phytoplankton and allochtonous
material were the major contributors (43-86 %) to the sediment surface Corg pool. In Denmark
*Z. marina* contributed with 13-81 % to the sediment surface Corg pool, contribution being





lowest at the most exposed site Nyborg and highest in Visby. The corresponding numbers for
Finland were 1.5-32 %, being lowest and highest in Tvärminne and Lyddaren, respectively (Fig.
5). The DistLm analysis showed that the *Z. marina* contribution to the sediment surface $^{13}$C pool
explained 10.9 % of the variation in the measured Corg stocks (Fig. 6, Table 3 and Table 4). Drift
algae was a significant contributor (72%) to the sediment surface Corg pool at the Danish sites,
while it appeared to play only a minor role (0-21%) in Finland. The carbon sources were
generally more mixed at the Danish study sites compared to the Finnish sites where
phytoplankton dominated (Fig. 5).

The $\delta^{13}$C values of the surface sediment within regions where quite homogenous ranging from -
18.9 to -22.8 ‰ and -13.5 to -17.6 ‰, in Finland and Denmark respectively. The $\delta^{13}$C in *Z.*
*marina* tissues ranged from -8.5 to -11.4 ‰ and from -8.2 to -12.5 ‰, in Finland and Denmark,
respectively. There was no significant difference between living above- and belowground tissue
and decomposed belowground tissue and samples were pooled in the isotope mixing model.
Although *Z. marina* was the dominant seagrass species in Finland, the study sites included both
monospecific and mixed seagrass meadows, where species like *Potamogeton pectinatus* and
*Potamogeton perfoliatus* were growing in mixed stands with *Z. marina*. *P. pectinatus* ($\delta^{13}$C -11.3
to -7.6 ‰) and *P. perfoliatus* ($\delta^{13}$C -15.6 to -12.6 ‰) were both present at five of the Finnish
study sites (Jänisholm, Sackholm, Hummelskär, Tvärminne and Fårö) and *P. pectinatus* was
present at Kolaviken, Ryssholmen and Lyddaren. *Ruppia cirrhosa* (-11.5 to -8.8 ‰) was less
abundant and found at three of the Finnish sites (Sackholm, Ängsö, Kolaviken) and at one study
site in Denmark (Kertinge). The $\delta^{13}$C for phytoplankton ranged from -24.6 to -22.6 ‰ and -18.6
to 16.4 ‰, in Finland and Denmark, respectively. Drift algae was present at all Danish study
sites, except Thurøbund, and had $\delta^{13}$C values from -17.9 to -13.5 ‰, but only at five Finnish sites
(Ängsö, Ryssholmen, Fårö, Långören and Hummelskär) with $\delta^{13}$C values ranging from -20.0 to -
16.3‰.




Discussion

Recent studies have shown considerable variation in the global estimates of carbon stocks and
carbon burial rates in seagrass meadows, indicating an incomplete understanding of factors
influencing this variability (Fourqurean et al. 2012, Duarte et al. 2013a, Lavery et al. 2013,
Miyayima et al. 2015). The Baltic Sea forms a key distribution area for eelgrass in Europe, but
similarly to the global data sets, we have so far lacked estimates on seagrass carbon stocks and
burial rates.

In our study, the Finnish eelgrass meadows showed consistently very low Corg stocks and Corg
accumulation, and the meadows were minor carbon sinks compared to the Danish meadows.
The Danish sites showed more variation in the sediment Corg stock and accumulation and Corg
stock was particularly high at one site, Thurøbund (26138 ±385 g C m$^{-2}$), which is a relatively
sheltered site with high organic sediments. Expectedly, due to both larger overall eelgrass
acreage and larger Corg stocks in the Danish meadows, the total Corg accumulation (0.38- 9.86 t
y$^{-1}$) was three to four orders of magnitude higher than in the Finnish meadows (0.002-0.033 t y$^{-1}$
$^{1}$). As eelgrass in Finland generally grow in more exposed locations potentially due to increased
interspecific competition with freshwater plants such as common reed (*Phragmites australis*) in
sheltered locations (Boström et al. 2006), it is probable that most of the Corg produced in the
Finnish meadows is exported, and thus incorporated in detrital food webs in deeper bottoms.
This argument is also supported when applying sedimentation rates from literature. Thus only
0.15 - 2% of the annual production was accumulated in Finnish meadows, while the
corresponding numbers for Denmark were 0.6 -7.8%. Duarte and Cebrian (1996) estimated that
on average 25% of the global seagrass primary production is exported, and seagrass detritus
may thus contribute significantly to Corg stocks in other locations, a fact that is often
overlooked.





Extrinsic drivers of carbon sequestration in seagrass meadows

The DistLm analysis showed, that three sediment variables (dry density, silt content, porosity)
and three plant variables (annual eelgrass production, the root:shoot-ratio and *Z. marina*
contribution to the sediment $^{13}$C pool) explained 67% of the variation in the sediment Corg stock
(g C m$^{-2}$) (Table 3 and 4, Fig. 6). Specifically, sediment silt content alone explained > 36 % of the
variation in Corg stocks (Table 3). In both regions, exposed sites characterized by sandy, low
organic sediments and low silt content, had low Corg stocks. In contrast, at sheltered sites like
Thurøbund in Denmark, we measured the highest sediment Corg stock along with highest silt
and water content among all sites. Although sediment porosity and sediment dry density also
contributed to the model, they were of minor importance (~2 % each. As proposed in previous
work (Kennedy et al. 2010, Miyajima et al. 2015) accumulation of fine grained size fractions in
seagrass sediments, relative to those accumulated in bare sediments, appears to be one of the
major factors influencing the carbon sink capacity of seagrass meadows, and may thus be a
useful proxy for the sink capacity.

In addition, it is well known, that seagrasses modify sediments by reducing water flow and
consequently increasing particle trapping and sedimentation and reducing resuspension
(Fonseca and Fisher 1986, Fonseca and Cahalan 1992, Gacia et al. 2002, Hendriks et al. 2008,
Boström et al. 2010) and also increasing Corg (Kennedy et al. 2010). Our finding of low carbon
sink capacity of Finnish seagrass meadows was supported by low sedimentation rates (3.6-7.7
gDW m$^{-2}$ d$^{-1}$) compared to the Danish sites (37.6-63.4 gDW m$^{-2}$ d$^{-1}$). These rates are similar to
sedimentation rates measured in previous studies (1.5 - 500 and 3.1- 20 gDW m$^{-2}$ d$^{-1}$; Gacia and
Duarte 2001, Holmer et al. 2004) from *P. oceanica* meadows. Thus, at the Finnish sites, the input
of organic particles and the potential for carbon accumulation and burial of eelgrass detritus and
external organic matter to the sediment is low.




Furthermore, the DistLm analysis showed, that *Z. marina* contribution to the sediment surface

carbon pool was an important driver (> 10.9%) of the variation in sediment Corg stock (Table 3

and 4, Fig. 6). We found increasing Corg stocks at the Danish sites, where *Z. marina* was the

major source of organic carbon, contributing with 13 to 81% to the sediment surface Corg. In

contrast, at the Finnish sites where major fraction of the carbon buried in the sediments derive

from phytoplankton and allochtonous material (>43%) the Corg stocks were low. In contrast,

the average $\delta^{13}C$ value (-16.2‰) in the Danish sediment samples was strikingly similar to the

global median value (−16.3‰±0.2‰) reported by Kennedy et al. (2010) in which on average 51

% of the carbon was derived from seagrass detritus compared to average 13-81 % for Danish

sites. The importance of the *Z. marina* contribution to the Corg stocks may be explained by slow

decomposition rates of seagrass tissue. Especially, the high proportion of refractory organic

compounds in the seagrass belowground parts and high C:N:P-ratios of seagrass tissue in

general make seagrasses less biodegradable than most marine plants and algae (Fourqurean and

Schrlau 2003, Vichkovitten and Holmer 2004, Kennedy and Björk 2009, Holmer et al. 2011). The

slow decomposition rates are also a result of reduced sediment conditions commonly

encountered in Danish seagrass meadows (Kristensen and Holmer 2001, Holmer et al. 2009,

Pedersen et al. 2011). Despite the extensive distribution (2-29 ha), high biomasses (300-800 g

DW m$^{-2}$) and major impact of drifting algal mats on coastal ecosystem functioning (Norkko &

Bonsdorff 1996, Salovius & Bonsdorff 2004,Rasmussen et al. 2013, Gustafsson & Boström 2014)

, the stable isotope composition of the sediments suggests that drift algae had a surprisingly

minor influence on the sediment surface Corg pool. Thus, despite present on several sampling

sites, seagrass detritus and drift algae is likely exported and mineralized in the water column

and in deeper sedimentation basins. Furthermore, we found that at all study sites in both

regions, there were several other potential sources influencing the sediment surface Corg pool.

Bouillon et al. (2007) showed that in seagrass sediments adjacent to mangrove forests in Kenya,

none of their sites had seagrass material as the sole source of Corg, instead mangrove-derived



detritus contributed significantly to the tropical seagrass sediment Corg pool. Similarly, at
majority of our study sites we observed several species of macroalgae and seagrasses that
contributed to the sediment Corg pool, although this contribution was of minor importance at
the Finnish sites.

The root: shoot-ratio explained 12.7 % of the variation in the Corg stocks. The highest Corg
stocks, below-ground biomass and root: shoot-ratio was found in Thurøbund (Denmark). The
relatively high explanatory value of the root: shoot-ratio could be explained by lower
decomposition rates of the *Z. marina* belowground tissue. In Finland, the highest root: shoot-
ratio (2.07) was found at Kolaviken, with a relatively low Corg stock (397 gDW C m$^{-2}$). Due to
higher degree of exposure at the site (degree of sorting 0.7 φ) compared to Thurøbund (1.4 φ) it
is likely that large portion of the eelgrass production was exported away from the meadow and
not stored in the sediment. The mean shoot densities were almost identical between regions,
and shoot density did not contribute to the model explaining Corg.

The annual eelgrass production explained only 2.3 % of the variation in the Corg stocks. The
annual production rates were almost twice as high at Danish sites compared to the Finnish sites.
Regional differences in seagrass productivity may be caused by differences in e.g. the inorganic
carbon concentration in water column and light availability between the regions (with higher
values in Denmark), which both affect the photosynthetic capacity of the plant (Hellblom and
Björk 1999, Holmer et al. 2009, Boström et al. 2014). Eelgrass production often tend to be higher
in physically exposed areas compared to more sheltered areas, which can be due to improved
sediment oxygen conditions and hydrodynamical effects (Hemminga and Duarte 2000). This
finding was not supported by our study, in which we found the highest annual eelgrass
production rates at both the most sheltered and exposed sites sites, namely Visby and Nyborg
(DK).



Carbon accumulation and stocks

Our estimated Corg stocks for the study sites were generally lower (627- 4360 t C km$^{-2}$) than
estimates (25200-84000 t C km$^{-2}$) found in the literature (Duarte et al. 2005, Nellemann et al.
2009, Mcleod et al. 2011, Fourqurean et al. 2012). In Duarte et al. (2005) the data set used for
the calculations was gathered from various studies conducted at different temporal scales and
habitat types, as well as different methods for determination of Corg accumulation. Additionally,
several of the studies were conducted in the Mediterranean *P. oceanica* meadows - a habitat with
exceptionally high carbon sequestration and storage capacity (Duarte et al. 2005, Lavery et al.
2013). In addition, the average sizes of Corg stocks in Finnish and Danish eelgrass meadows
were 6900 and 4320 t C km$^{-2}$, respectively, and thus also considerably lower than the mean
values reported by Alongi et al. (2014) for tropical seagrass meadows (14270 t C km$^{-2}$),
mangroves (95600 t C km$^{-2}$) and salt marshes (59300 t C km$^{-2}$). In contrast, our estimate for the
carbon stock in the top 25 cm for Danish and Finnish meadows (627-6005 g Corg m$^{-2}$) are
comparable to Australian (262–4833 g Corg m$^{-2}$: Lavery et al. 2013) and Asian estimates (3800-
12000 g Corg m$^{-2}$: Miyajima et al. 2015).

Consequences of seagrass loss for carbon pools

Despite the importance of seagrasses, their global distribution has decreased by 29% since 1879
primarily due to anthropogenic pressures (Waycott et al. 2009), thus weakening the carbon sink
capacity of marine environments to sequester carbon (Duarte et al. 2005). Since the 1970s, the
Baltic Sea has been subject to strong anthropogenic pressures (Conley et al. 2009) leading to
eelgrass declines in several countries (Boström et al. 2014). In the 1930s, the Danish eelgrass
meadows were significantly reduced by the wasting disease (Rasmussen 1977). These regime
shifts in Denmark, have resulted in a 80-90 % decline corresponding to 6726 km$^{2}$ in the
beginning of 1900`s to 673-1345 km$^{2}$ in 2005, using the minimum and maximum estimates for



the current coverage area, respectively (Boström et al. 2014). Similarly, eelgrass meadows in
Sweden have declined by some 60 % since the mid-1980s resulting in a present coverage of 68
(minimum) to 138 (maximum) km². In Germany the eelgrass coverage area has decreased with
75 %, resulting in the present eelgrass area of 147 km² (Boström et al. 2014). In Finland there is
a lack of long-term monitoring, but the meadows appear to be stable and cover at least 30 km². It
is clear, that these large-scale seagrass declines have eroded the Corg stocks in the Baltic Sea
significantly (Table 2). Using the mean Corg (17.45 mg C cm⁻³) measured at the Danish sites, the
loss in Corg storage capacity is estimated to 0.4-0.9, 23-27 and 1.9 Mt Corg in Sweden, Denmark
and Germany, respectively.

For the Swedish west coast, Moksnes and Cole (2015) estimated an annual economic loss due to
the lost seagrass carbon fixation capacity to be 248 € ha⁻¹y⁻¹ (carbon price 117 € t). If also the
carbon stored in the top 25 cm of sediment, as well as the loss of seagrass carbon sequestration
capacity over 50 year period were taken into account, the value of the lost carbon storage and
sequestration capacity was approximately 5321 € ha⁻¹. This value is slightly higher than our
estimates for the monetary value of the present carbon storage and sequestration capacity
eelgrass meadows at Finland and Denmark (346 and 1862 € ha⁻¹). This difference is mainly due
to the lower (40.3€) monetary value of carbon used in our calculations. Pendleton et al. (2014),
calculated a global estimated economic cost of lost seagrass meadows (CO₂ price 41 $ t) to be
1.9-13.7 billion USD. This value was derived only from the cost of the lost carbon sink capacity,
ignoring the other lost ecosystem services including e.g. coastal protection, water quality
management, food provision and the role of seagrasses as fisheries and key habitats for marine
species (Barbier et al. 2011, Atwood et al. 2015). Thus, we estimate the present economic value
of carbon storage and sequestration capacity of Baltic Sea and Norwegian eelgrass meadows to
be between 1.7 and 12 % of the global seagrass blue carbon value.



While useful, our and previous work still remain snap shots of complex processes causing local
and regional variability in estimates of seagrass Blue Carbon stocks and accumulation. Clearly, in
order to produce more reliable estimates of global seagrass carbon burial rates and stocks, there
is a need for more studies integrating and modeling the individual and joint role of e.g. sediment
biogeochemistry, seascape structure, plant species architecture and hydrodynamic regime. Since
seagrasses are lost at accelerating rates (Waycott et al. 2009), there is also an urgent need for a
better understanding of the fate of lost seagrass carbon (Macreadie et al. 2014) and the
development of the carbon sink capacity in restored seagrass ecosystems (Nellemann et al.
2009, Greiner et al. 2013, Marba et al. 2015). Nelleman et al. (2009) proposed the use of carbon
trading programs using financial incentives for forest conservation, such as REDD+ (Reduced
Emissions from Deforestation and Degradation) and NAMAs (Nationally Appropriate Mitigation
Actions), to include the blue carbon ecosystems as part of their environmental protection
protocol. Both of these carbon mitigation programs require ongoing monitoring of organic
carbon storage and emission in the different Blue Carbon ecosystems. In order to manage
seagrass meadows, mitigate climate change and produce information acquired for the carbon
trading programs, it is fundamental to understand factors influencing the capacity of seagrass
meadows to capture and store carbon. By solving these uncertainties, the conservation and
restoration of seagrass meadows can be implemented in the most beneficial manner by e.g.
giving priority to protection of the seagrass meadows and species with the highest carbon sink
capacity and foundation of restoration projects in areas most suitable for seagrass growth
(Duarte et al. 2013a).



Copyright statement






Acknowledgements

This work is part of a double degree program between Åbo Akademi University (ÅAU) and
University of Southern Denmark (SDU). The study was funded by The Maj and Tor Nesslings
Foundation and University of Southern Denmark. We are grateful for Archipelago Centre
Korpoström and University of Southern Denmark for excellent working facilities. Tiina Salo and
Karine Gagnon are acknowledged for field assistance and Katrine Kierkegaard and Birthe
Christensen for their assistance in the laboratory work.












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



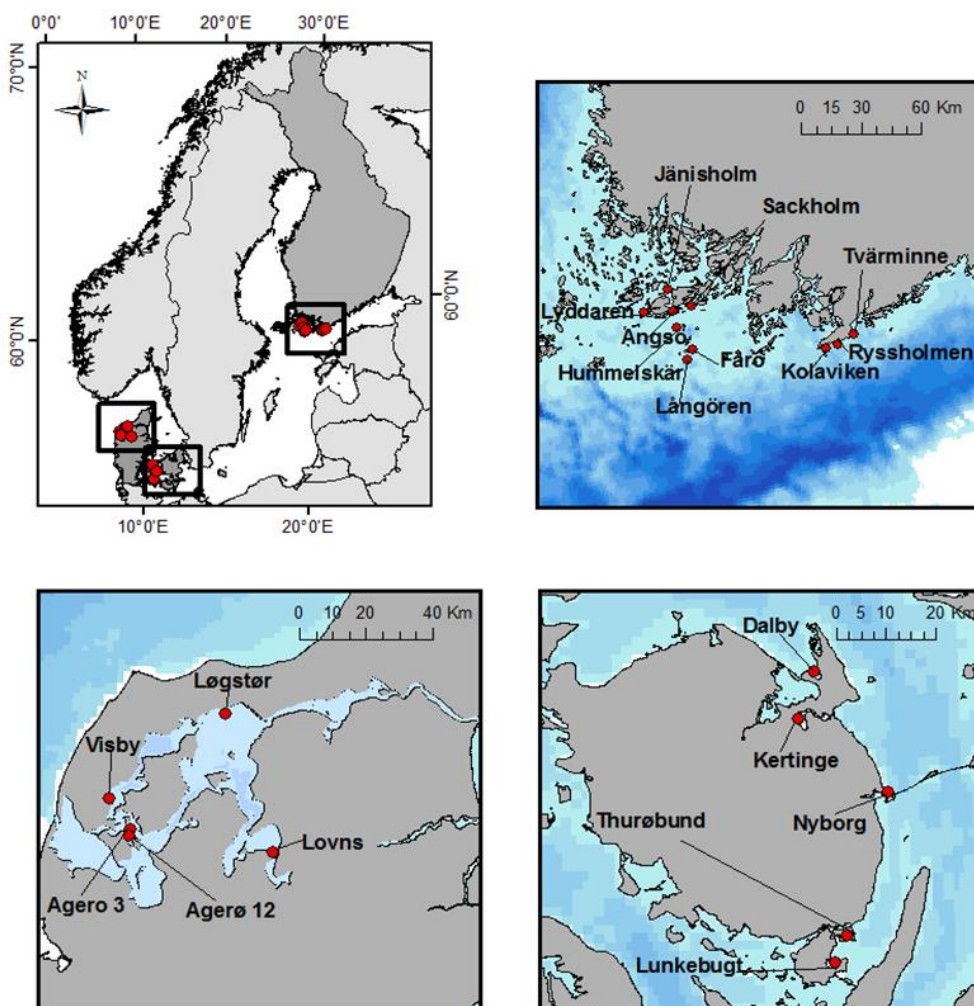

Fig.1. The study sites in Denmark and Finland.



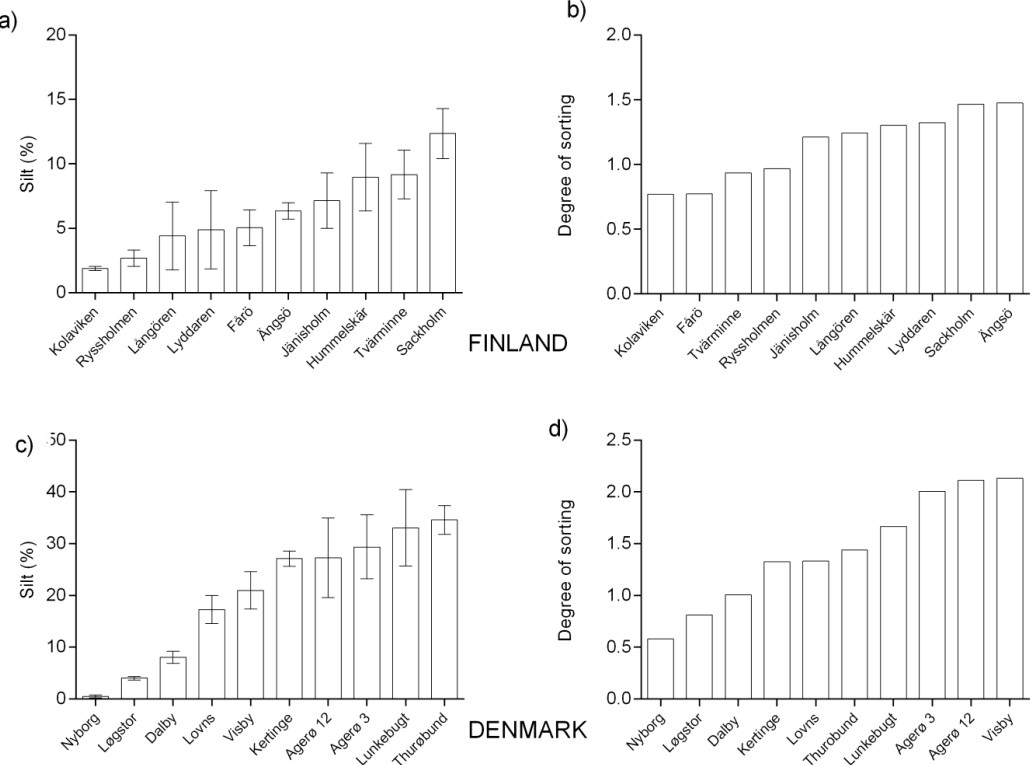

Fig. 2. A percentage silt (a, c) and degree of sediment sorting (b, d) at the study sites in Finland
and Denmark, respectively. Lower values in degree of sorting indicate well-sorted sediment
types.





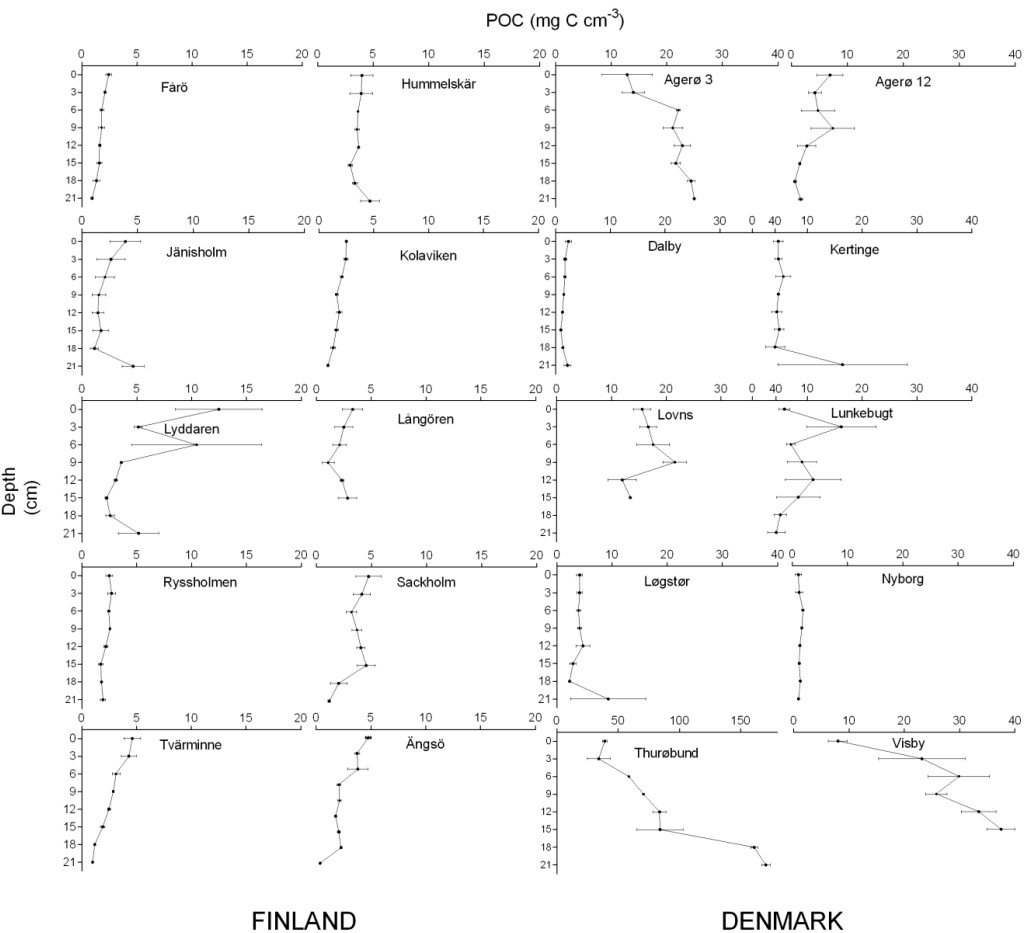

Fig.3. Sediment profiles of particulate organic carbon (POC) content (mg C cm$^{-3}$) in the top 25 cm
of the Finnish and Danish eelgrass (*Zostera marina*) meadows. Note the difference in the scale of
x-axis between the regions.






Fig.4. Organic carbon stocks (Corg, g C m$^{-2}$) in the top 25 cm of sediment in Finnish and Danish
eelgrass (*Zostera marina*) meadows. Note Thurøbund (grey column) as the single site belonging
to right y- axis.
















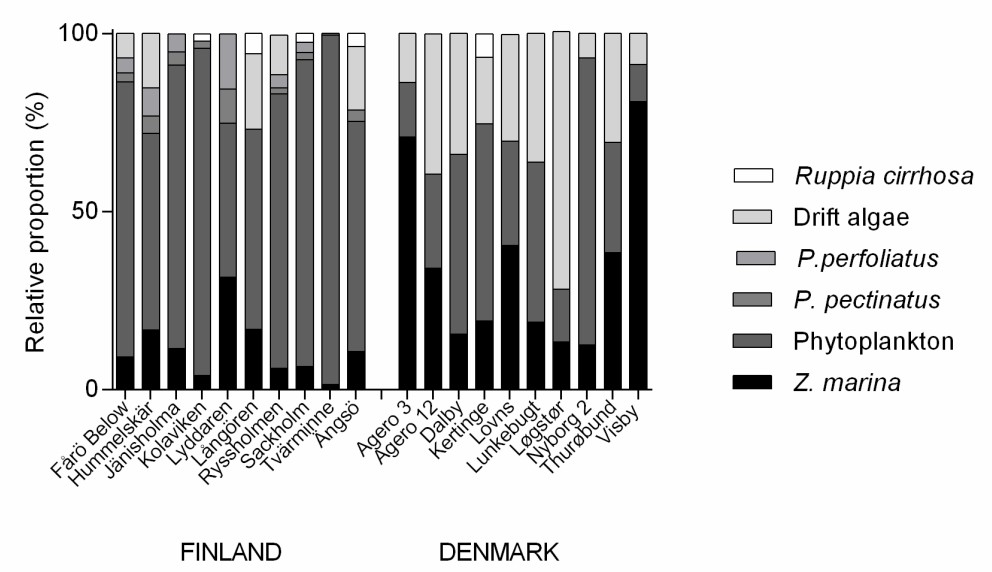


Fig.5. Relative contribution of organic matter sources (*Z. marina, P. perfoliatus, P. pectinatus,*
*Ruppia cirrhosa,* phytoplankton and drift algae) to the $^{13}$C signal of the sediment surface layer (0-
2 cm) in Finnish and Danish eelgrass (*Zostera marina*) meadows.



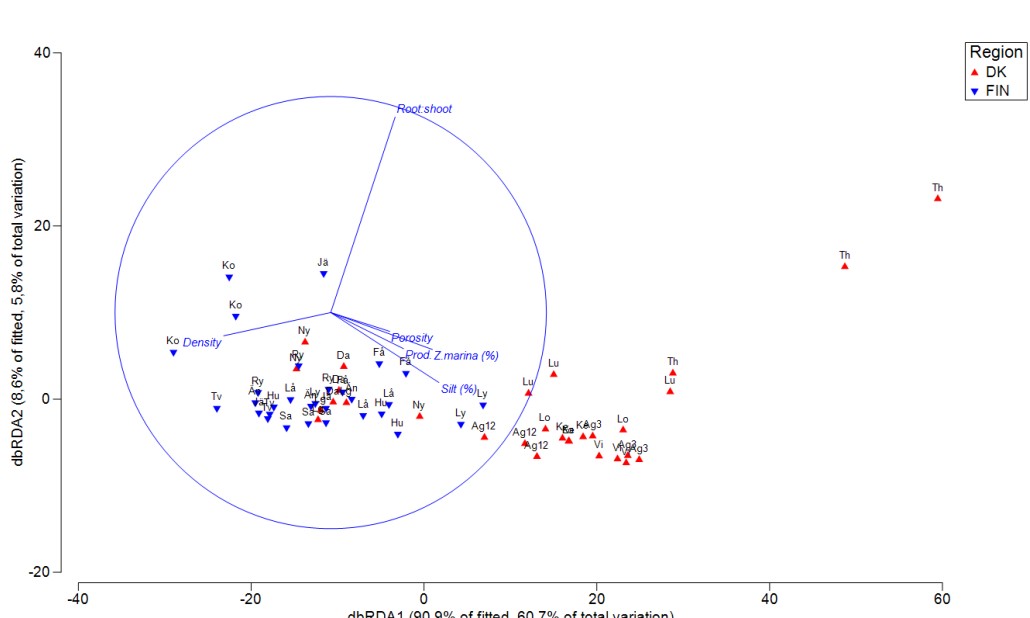

Fig.6. Distance-based redundancy analysis (DbRDA) plot showing the environmental parameters (percentage of *Z. marina* in sediment carbon pool, above: belowground- ratio, annual eelgrass production, sediment silt content, sediment dry density and sediment porosity) fitted to the variation in the Corg stock (g C m⁻²) at the Finnish and Danish eelgrass (*Zostera marina*) sites, respectively. Vectors indicate direction of the parameters effect. Site codes: Finland; Ko=Kolaviken, Ry=Ryssholmen, Tv=Tvärminne, Få=Fårö, Ly=Lyddaren, Lå=Långören, Hu=Hummelskär, Jä=Jänisholm. Site codes: Denmark; Ag12=Agerø 12, Ag3=Agerø 3, Vi=Visby, Lg=Løgstør, Lo=Lovns, Th=Thurøbund, Lu=Lunkebugt, Da=Dalby, Ke=Kertinge, Ny=Nyborg.





Table 1. Location, silt content (% silt), sediment organic matter content (%DW), seagrass shoot
density (shoots m$^{-2}$), seagrass above and below-ground biomass (gDW m$^{-2}$), root: shoot-ratio,
and above-ground production (gDW m$^{-2}$ y$^{-1}$) at the sampling sites. SE (n= 3–4) is given. Annual
seagrass production is calculated from pooled values of replicates per site and therefore no SE is
shown.

| Location | Silt content (% silt) | Organic matter content (% DW) | Shoot density (shoots m$^{-2}$) | Above-ground biomass (gDW m$^{-2}$) | Below-ground biomass (gDW m$^{-2}$) | Root: shoot-ratio |
|---|---|---|---|---|---|---|
| Finland | | | | | | |
| Fårö | 5.0±1.4 | 0.66±0.07 | 304±32 | 138±20 | 167±28 | 1.27±0.13 |
| Hummelskär | 9.0±2.6 | 1.06±0.20 | 364±31 | 70±11 | 28±2 | 0.45±0.06 |
| Jänisholm | 7.1±2.1 | 0.93±0.20 | 128±17 | 65±16 | 46±2 | 1.44±0.53 |
| Kolaviken | 1.9±0.2 | 0.75±0.02 | 476±96 | 74±6 | 149±16 | 2.07±0.27 |
| Lyddaren | 4.9±2.5 | 1.75±0.70 | 228±42 | 86±7 | 57±12 | 0.64±0.09 |
| Långören | 4.4±2.1 | 2.70±2.10 | 436±53 | 121±46 | 68±25 | 0.58±0.06 |
| Ryssholmen | 2.7±0.6 | 0.89±0.20 | 756±57 | 160±3 | 136±16 | 0.86±0.11 |
| Sackholm | 12.4±1.9 | 0.95±0.20 | 774±234 | 110±18 | 37±9 | 0.31±0.04 |
| Tvärminne | 9.2±1.9 | 0.88±0.20 | 112±11 | 99±16 | 38±7 | 0.37±0.01 |
| Ängsö | 6.3±0.5 | 0.84±0.02 | 604±98 | 91±6 | 63±9 | 0.67±0.05 |
| **Finland average** | **6.3±1** | **1.4±0.3** | **417±75** | **101±3** | **79±5** | **0.87±0.06** |
| Denmark | | | | | | |
| Agero 3 | 29.4±6.2 | 1.94±0.60 | 448±89 | 181±33 | 84±8 | 0.52±0.07 |
| Agero 12 | 27.3±7.7 | 1.65±0.80 | 404±90 | 110±2 | 46±9 | 0.40±0.08 |
| Dalby | 8.1±1.2 | 0.67±0.03 | 400±48 | 76±7 | 83±10 | 1.09±0.11 |
| Kertinge | 27.1±1.5 | 12.59±1.60 | 328±64 | 90±17 | 64±14 | 0.68±0.02 |
| Lovns | 17.3±2.7 | 2.90±0.50 | 360±27 | 141±4 | 100±11 | 0.70±0.06 |
| Lunkebugt | 33.0±7.4 | 4.72±2.40 | 347±81 | 210±10 | 382±24 | 1.82±0.08 |
| Løgstør | 4.0±0.4 | 0.75±0.03 | 300±14 | 149±11 | 63±13 | 0.42±0.07 |
| Nyborg | 0.5±0.3 | 0.42±0.02 | 652±30 | 203±24 | 214±50 | 1.00±0.14 |
| Thurøbund | 34.6±2.8 | 14.48±0.80 | 420±98 | 101±16 | 398±15 | 4.54±0.70 |
| Visby | 21.0±3.6 | 1.17±0.06 | 520±21 | 193±13 | 49±4 | 0.25±0.01 |
| **Denmark average** | **20.2±3.9** | **3.9±1.5** | **418±32** | **145±5** | **148±14** | **1.14±0.13** |




Table 2. Estimated average carbon stocks (g C m$^{-2}$ and Mt) and annual carbon accumulation
(Annual Corg (Mt y$^{-1}$) in Finnish and Danish eelgrass (*Zostera marina*) meadows. Denmark$_{lost}$ =
eelgrass area of the region lost since the beginning 1900`s. Limfjorden$_{lost}$= eelgrass area of the
region lost since the 1900`s. See text for calculations. *) mean Corg (mg C cm$^{-3}$) calculated for
Denmark is used. n.d= no data. For calculations of annual carbon accumulation three different
sedimentation rates were applied (0.32 mm y$^{-1}$; Miyayima et al. 2015, 2.02 mm y$^{-1}$; Duarte et al.
2013b and 4.2 mm y$^{-1}$; Serrano et al. 2014).





| Region | Seagrass area (km²) | Corg stock (mg C cm$^{-3}$) | Corg stock (g C m$^{-2}$) | Corg stock (t) | Annual Corg ( Mt y$^{-1}$) | | |
|---|---|---|---|---|---|---|---|
| | | | | | 0.32 mm y$^{-1}$ | 2.02 mm y$^{-1}$ | 4.20 mm y$^{-1}$ |
| Finland | 30 | 2.60±0.09 | 627±25 | 0.019±< 0.001 | 0.002 | 0.016 | 0.0328 |
| Limjorden | 18 | 10.57±1.66 | 2644±207 | 0.047± 0.007 | 0.006 | 0.038 | 0.079 |
| Funen | 179 | 24.32±9.15 | 6005±1127 | 1.090±0.410 | 0.139 | 0.881 | 1.832 |
| Denmark$_{min}$ | 673 | 17.45±9.42* | 4324±1188* | 2.164±0.005 | 0.376 | 2.373 | 3.636 |
| Denmark$_{max}$ | 1345 | 17.45±9.42* | 4324±1188* | 5.868±0.014 | 0.75 | 4.741 | 9.859 |
| Denmark$_{lost}$ | 5381-6230 | 17.45±9.42* | 17.45±9.42* | 23.478-27.183 | n.d | n.d | n.d |












Table 3. Table from DistLm analysis showing variables in the marginal tests and the results for
statistical analysis.

**MARGINAL TESTS**

| Variable | Sum of Squares | Pseudo-F | P-value | Proportion |
|---|---|---|---|---|
| 1. Root: shoot- ratio | 5309 | 10.64 | 0.002 | 0.155 |
| 2. Sediment dry density | 10704 | 26.37 | 0.001 | 0.313 |
| 3. Annual eelgrass production | 4959 | 9.82 | 0.002 | 0.145 |
| 4. Shoot density | 48 | 0.08 | 0.911 | 0.001 |
| 5. Porosity | 3507 | 6.61 | 0.01 | 0.102 |
| 6. % silt | 12653 | 33.99 | 0.001 | 0.369 |
| 7. C:N-ratio of plant material | 464 | 0.79 | 0.397 | 0.014 |
| *8. Z. marina %* | 12179 | 32.02 | 0.001 | 0.356 |
| 9. Degree of sorting | 9725 | 23.01 | 0.001 | 0.284 |





















Table 4. Table from DistLm analysis showing results from the sequential tests and solution given
by the analysis.

| Variable | AICc | Sum of squares | Pseudo-F | P- value | Proportion | Cumulative proportion | Degrees of freedom |
|---|---|---|---|---|---|---|---|
| 6. % silt | 357.4 | 12653 | 33.9 | 0.001 | 0.369 | 0.369 | 58 |
| 1. Root :shoot-ratio | 346.0 | 4375 | 14.5 | 0.001 | 0.127 | 0.497 | 57 |
| 8. *Z. marina* % | 333.6 | 3745 | 15.6 | 0.001 | 0.109 | 0.606 | 56 |
| 3. Production | 332.2 | 805 | 3.5 | 0.037 | 0.023 | 0.630 | 55 |
| 2. Density | 331.3 | 700 | 3.2 | 0.049 | 0.020 | 0.650 | 54 |
| 5. Porosity | 330.8 | 602 | 2.8 | 0.056 | 0.017 | 0.668 | 53 |
| BEST SOLUTION | AICc | R^2 | RSS | Variables | Selections | | |
| | 330.8 | 0.668 | 11363 | 6 | 1-3;5;6;8 | | |











