# Peer review of "Blue carbon stocks in Baltic Sea eelgrass (*Zostera marina*) meadows"

_Biogeosciences, 2016_

## Referee Comment (RC1) · Anonymous Referee #1 · 8 Jun 2016

The manuscript by Röhr et al. presents a potentially interesting dataset on sediment organic carbon stocks and sedimentation rates in constrasting seagrass meadows in the Baltic Sea region. At this stage, however, I cannot recommend publication as there are a number of fundamental flaws in methodology and presentation of the results which compromise the use and interpretation of the data. I have recommended rejection to indicate that it needs a very thorough overhaul; although I feel that with proper re-analysis of (samples and) data, this could certainly be resubmitted.

My main concerns are outlined below.

-The authors mention that sediments were not acidified to remove carbonates as the inorganic carbon content in their samples was low but they indicate a range of up to 0.3 % inorganic C. I do not understand why a simple acidification procedure was not

[Figure]

followed for all samples – if some samples contain 0.3% inorganic C this could substantially influence the d13C values measured, since OC concentrations are often also low. Table 1 shows organic matter contents (is this really %OM, or %OC ? This should also be clarified) and the averages are as low as 0.42 % for some sites. If this is %OC (and even worse, if it is %OM), that would imply that up to half the C measured could be inorganic carbon, and this would severely bias the OC stock numbers as well as the d13C data. Acidifying these samples after weighing them takes a few minutes – and avoids any concerns on bias due to inorganic C contributions- hence I do not understand why one would not due this routinely on all samples. The authors should either show more convincing data that the data are not biased by inorganic C contributions, or re-analyze their samples with proper acidification to remove carbonates. All d13C data should also be presented somehow in Tables or Figures and not merely in the text as is currently the case.

-The authors provide estimates for Corg accumulation but it's very unclear what these actually represent and how they should be intepreted. In their study, they measured sedimentation rates using sediment traps (hence, the downward flux of suspended matter and OC) but this is not the same as OC accumulation rates, since much of the OC reaching the surface is likely to be mineralized rather than buried. %OC and d13C data should be clearly shown for but the sediment trap samples and sediments profiles. On L225-230 the authors explain how "Corg accumulation" was calculated: Results for carbon burial (applied by multiplying the Corg stock, regional seagrass area and sedimentation rate estimate from literature) in each area are given as Corg accumulation (t y-1)". This does not make sense, since (i) this formula does not match in terms of units (Corg stock: g C m-2; area: m2; sedimentation rate: g DW m-2 y-1; so you end up with gC x g DW m-2), (ii) why use sedimentation rates from the literature when you measured these with your sediment traps ? The literature sources referred to here for sedimentation rates are not from these sites, hence they are unlikely to be very meaningful. In order to obtain Corg accumulation rates for your study sites, the %OC profiles should be combined with sediment dating techniques (e.g. 210Pbxs) or

site-specific sedimentation rates.

-The authors mention a few times that sediment density is a factor that partially explains the OC stocks. This is not a causal factor – density and %OC are expected to be inversely related in sediments since organic matter has a much lower density than the mineral fraction. See for example the supplementary information in Donato et al. (2011) Mangroves among the most carbon-rich forests in the tropics. Nature Geoscience, doi: 10.1038/ngeo1123. http://www.nature.com/ngeo/journal/v4/n5/extref/ngeo1123-s1.pdf

-L233: be more specific in terminology: how do you define "eelgrass carbon sequestration rate": carbon burial ? net primary production ? Why do you need this (a rate) to calculate the total Corg pool (= a stock) ?

-L326: how can the Corg stock be as low as <0.001 mg C cm-3 ? Please check these numbers. You refer to Figure 3 here but I don't see these low numbers on Figure 3.

Minor comments: -L17: account –>→ accounts

-L17: oceanic carbon burial : organic ? total ?

-L19: organic carbon

-L20: accumulation rates

-L24: organic carbon

-L27-28: these should be expressed in areal rates (or in areal rates + integrated over the relevant areas)

-L29-30: see comment above regarding density

-L31: DistLm: explain

-L44: Atmospheric carbon dioxide

-L59: 10ˆ-10 : something wrong here. Express this is e.g. PgC

-L66 and further: salinity has no units – avoid using psu

-L73-74: 7% decline per year – but a total 29% loss since 1879; these numbers are indeed often cited but they are somewhat at odds – if the 7% per year is consistent, that would lead to >29% loss in 4 years.

-L96: "typically faster": ambiguous: do you mean that seagrasses have faster turnover rates, or other sources ?

-L98: use the correct terminology: "isotopically heavier in d13C" is not correct: "enriched in 13C", or "have higher d13C values"

–L103-106: explain how identifying/constraining sources can lead to more reliable estimates of the capacity of the meadows to store and sequester carbon. I don't think this is the case – they allow you to budget the fate of seagrass C more accurately, but to determine the organic C stores or sequestration rates, you don't need the source information ?

-L333: "ranged from": but then you provide an average +/- stdev.

-L385 and elsewhere: when presenting a range of d13C values, provide the lowest (most negative value) first.

---

## Author Comment (AC1) · 22 Jun 2016

June 22 2016

Dear Editor,

Please find our detailed responses to the individual questions posed. We found the referee comments very useful for both correcting details and improving the overall clarity of the manuscript. I have attached a corrected version of the ms as a supplement to this comment, and changes to Tables 1 and 2 can be seen in the supplement. Looking forward to your response,

Sincerely, Emilia Röhr

The manuscript by Röhr et al. presents a potentially interesting dataset on sediment organic carbon stocks and sedimentation rates in contrasting seagrass meadows in the Baltic Sea region. At this stage, however, I cannot recommend publication as there are a number of fundamental flaws in methodology and presentation of the results which compromise the use and interpretation of the data. I have recommended rejection to indicate that it needs a very thorough overhaul; although I feel that with proper re-analysis of (samples and) data, this could certainly be re-submitted.

My main concerns are outlined below.

1. The authors mention that sediments were not acidified to remove carbonates as the inorganic carbon content in their samples was low but they indicate a range of up to 0.3 % inorganic C. I do not understand why a simple acidification procedure was not followed for all samples – if some samples contain 0.3% inorganic C this could substantially influence the d13C values measured, since OC concentrations are often also low.

Reply: The acidification procedure is not trivial for the sediments examined here. Most of the sediments have, as mentioned by the reviewer, very low OC contents and we need to weigh maximum amount of sediment into the tin capsules making the acidification process difficult. Our approach was therefore instead to measure the inorganic carbon content at the 20 study sites to evaluate if IC was contributing significantly to TC in the sediments. This was not the case, as the IC content was between 0.5- 5% of the TC in the sediments. The error by including IC was considered to be minor considering the heterogeneity in the sediments. Further analysis of 13C signal in IC showed that the value was similar to the TC values within the natural heterogeneity of the sediments. Furthermore, as shown by Schlacher and Conolly (2014) acidification procedure should be used with caution as it could also significantly alter the 13 C signal, in particular in low inorganic carbon sediments such as the sediments in our study area. We prioritized to have 3 replicates of all sediment samples, which allow evaluation of the heterogeneity of the samples, which is often neglected in this type of studies due to the cost of analysis. Accordingly, we have corrected the text in the MS

by referring to IC content of TC (0.5-5 %), rather than providing the absolute values of IC (0.03-0.3%DW).

2. Table 1 shows organic matter contents (is this really %OM, or %OC ? This should also be clarified) and the averages are as low as 0.42 % for some sites. If this is %OC (and even worse, if it is %OM), that would imply that up to half the C measured could be inorganic carbon HOW DO YOU ARRIVE AT 50%??, and this would severely bias the OC stock numbers as well as the d13C data. Acidifying these samples after weighing them takes a few minutes – and avoids any concerns on bias due to inorganic C contributions- hence I do not understand why one would not due this routinely on all samples. The authors should either show more convincing data that the data are not biased by inorganic C contributions, or re-analyze their samples with proper acidification to remove carbonates.

Reply:It is OM % in the tables, which has been further clarified. See answer above regarding the procedures.

3. All d13C data should also be presented somehow in Tables or Figures and not merely in the text as is currently the case.

Reply:The d13 C data of sediment surface, Z. marina leaves and rhizomes has been added to table 1.

4. The authors provide estimates for Corg accumulation but it's very unclear what these actually represent and how they should be intepreted. In their study, they measured sedimentation rates using sediment traps (hence, the downward flux of suspended matter and OC) but this is not the same as OC accumulation rates, since much of the OC reaching the surface is likely to be mineralized rather than buried.

Reply:We fully agree with the reviewer and to address this issue we did not use the sedimentation rates we measured ourselves in the Corg accumulation calculations, instead we used the sediment accumulation rates from literature, that have been obtained from either 210Pb analysis, radiocarbon dating or long term monitoring (see answer for question 6 and below). Using published peer-reviewed values for sedimentation was the only possibility here (as in several other papers, e.g. Duarte et al. 2013, Lavery et al. 2013) as obtaining 210Pb cores was beyond the scope of this investigation.

5.%OC and d13C data should be clearly shown for the sediment trap samples and sediments profiles.

Reply:We have presented the POC depth profiles for all our study sites (Fig. 3). We found it uninformative to present the d13C depth profiles for all of the study sites. This is because we analyzed d13 C depth profiles for 8/10 Danish sites and saw no significant change in the isotope signal with depth. For that reason, and due to costs of analysis, we decided not to analyze the d13 C depth profiles from the remaining sites.Instead our focus is on comparing changes of d13 C signal from the sediment surface layer.

6. On L225-230 the authors explain how "Corg accumulation" was calculated: Results for carbon burial (applied by multiplying the Corg stock, regional seagrass area and sedimentation rate estimate from literature) in each area are given as Corg accumulation (t y-1)". This does not make sense, since (i) this formula does not match in terms of units (Corg stock: g C m-2; area: m2; sedimentation rate: g DW m-2 y-1; so you end up with gC x g DW m-2)

Reply:We have clarified this part in the MS. We used sediment accumulation rates from the literature that were presented in mm y-1, not our own sedimentation rates that were expressed in gDW m-2 y-1,and only represent a snapshot of the sedimentation and not the sediment accumulation rate (see further details below).

7.Why use sedimentation rates from the literature when you measured these with your sediment traps?

Reply:Sediment traps were deployed for short period of time (48 hours), and represent

temporally a snapshot of the sedimentation and not the actual sediment accumulation rates. The sediment traps were used for comparison of sedimentation rates between Finland and Denmark, as well as to evaluate the input sources of material to the sediments through the sedimentation process.

The sediment accumulation rates from literature were either based on 210Pb profiles, radiocarbon dating or long-term sediment trap deployments (Duarte et al. 2013,Serrano et al. 2014, Miyayima et al. 2015). The accumulation rates applied here vary from 0.32 mm y-1 to 4.2 mm y-1. Notably these numbers are very conservative estimates for sediment accumulation in coastal areas as higher rates can be expected.

8.The literature sources referred to here for sedimentation rates are not from these sites, hence they are unlikely to be very meaningful.

Reply:See previous answer. In addition, there are very few publications (Duarte et al. 2005, Fourqurean et al. 2012, Lavery et al. 2013, Geiner et al. 2013, Serrano et al. 2014a, Serrano et al. 2014b, Miyayima et al. 2015) that have reported sedimentation rates/ sediment accumulation rates in seagrass meadows and even less for Zostera marina on the northern hemisphere (Greiner et al. 2013, Miyayima et al. 2015). We made multiple literature searches and reviewed a number of papers, and are confident that the values used in our context (0.32, 2.02 and 4.2 mm y-1;Duarte et al. 2013,Serrano et al. 2014, Miyayima et al. 2015) are relevant.

As sediment accumulation data is generally very sparse, using literature data is often the only option for providing estimates for Corg accumulation in seagrass ecosystems. A similar approach to estimating Corg has been used in several influential research papers (e.g. Duarte et al. 2005 (cited 500 times), Lavery et al. 2013 (cited 50 times).

9. In order to obtain Corg accumulation rates for your study sites, the %OC profiles should be combined with sediment dating techniques (e.g. 210Pb) or site-specific sedimentation rates.

Reply:See answers above.

10.Authors mention a few times that sediment density is a factor that partially explains the OC stocks. This is not a causal factor – density and %OC are expected to be inversely related in sediments since organic matter has a much lower density than the mineral fraction. See for example the supplementary information in Donato et al. (2011) Mangroves among the mostcarbon-rich forests in the tropics. Nature Geoscience, doi:10.1038/ngeo1123.http://www.nature.com/ngeo/journal/v4/n5/extref/ngeo1123-s1.pdf

Reply:We do not understand this comment, as our data showed exactly this: high organic matter content sediments had the lowest sediment dry density as sediment dry density decreased with increasing OM content.

11.L233: be more specific in terminology: how do you define "eelgrass carbon sequestration rate": carbon burial ? net primary production ?

Reply:To clarify this, we have now added a definition for carbon sink capacity in line 57 and for carbon sequestrtaion in line 60.

12. Why do you need this (a rate) to calculate the total Corg pool (= a stock)?

Reply:We have changed this part in the ms and used Corg sequestration estimate calculated from our own data. In the calculation the Corg sequestration estimate is added to the POC we measured from above- and belowground biomass of Z.marina and the mean Corg of sediments to obtain a value for total Corg in the study area. Calculation is shown below. Carbon components for calculation of value or price:

DENMARK FINLAND 1. Annual C sequestration 0.35 ton/ ha 0.05 ton/ha Lost C sequestration each year over 100 yrs in Denmark 0.35 ton/ha x 100 = 35 t/ha

2. C in living eelgrass Denmark: Finland: Mean proportion aboveground POC in Zm 35% 38% Mean Proportion belowground POC in Zm 29% 36% Average aboveground

biomass 145 gDW m-2 101 gDW m-2 Average below ground biomass 148 gDWm-2 79 Proportion POC aboveground 50.7 g m-2 = 0.51 t/ha 38.4 g m-2 = 0.38 t/ha Proportion POC belowground 43 g m-2 =0.43 t/ha 28.4 gm-2= 0.28 t/ha Tot. abgr + belgr POC in living Zm 0.94 t/ha 0.66 t/ha 3. C in sediments Corg mean of n=10 sites 43.6 t/ha 6.27

TotC DEN= 0.35+0.94+43.6= 44.9 t/ha or 4490 t/km2 TotC FIN= 0.05+0.66+6.27= 6.98 t/ha or 698 t/km2

13. L326: how can the Corg stock be as low as <0.001 mg C cm-3 ? Please check these numbers. You refer to Figure 3 here but I don't see these low numbers on Figure 3.

Reply:These values are correct, but represent the mean from each site with standard error of mean. Many sites had replicates with POC (mg C cm-3) concentration below detection limit (0.001).

Minor comments:

14.L17: account –>accounts Reply:Corrected.

15. L17: oceanic carbon burial : organic ? total ? Reply:Corrected.

16.L19: organic carbon Reply:Corrected.

17. L20: accumulation rates Reply:Corrected.

18.L24: organic carbon Reply:Corrected.

19. L27-28: these should be expressed in areal rates (or in areal rates + integrated over the relevant areas)

Reply:The lines the reviewer is referring to (L 24-25) are areal estimates. A conservative estimate of the total organic carbon pool in the regions ranged between 6.98-44.9 t ha-1. Our results suggest that the Finnish eelgrass meadows are minor carbon sinks compared to the Danish meadows, and that majority of the Corg produced in the

Finnish meadows is exported. Similarly, the estimates for Corg accumulation in eelgrass meadows in Finland (< 0.002- 0.033 t C y-1) were over two orders of magnitude lower compared to Denmark (0.376-3.636 Corg t y-1).We have added a column to Table 2, in which we present the Corg sequestration rates calculated individually for the different study regions.

20. L29-30: see comment above regarding density

Reply:See answer 10.

21. L31: DistLm: explain

Reply:Explanation added.

22. L44: Atmospheric carbon dioxide

Reply:Corrected.

23. L59: 10ËĘ-10 : something wrong here. Express this is e.g. PgC

Reply:Corrected.

24.L66 and further: salinity has no units – avoid using psu

Reply:Corrected.

25.L73-74: 7% decline per year – but a total 29% loss since 1879; these numbers are indeed often cited but they are somewhat at odds – if the 7% per year is consistent, that would lead to >29% loss in 4 years.

Reply:These numbers are directly from Waycott et al. PNAS. 29% of the known global areal extent has disappeared since seagrass areas were initially recorded in 1879. Rates of decline have accelerated from a median of 0.9% yr-1 before 1940 to 7% yr-1 since 1990. According to one of the authors, this is calculated as 7% of the total remaining at the end of each year, which means that the actual amount of seagrass area lost decreases each year because the total decreases each year.

26.L96: "typically faster": ambiguous: do you mean that seagrasses have faster turnover rates, or other sources ?

Reply:Corrected.

27.L98: use the correct terminology: "isotopically heavier in d13C" is not correct: "enriched in 13C", or "have higher d13C values"

Reply:Corrected.

28.L103-106: explain how identifying/constraining sources can lead to more reliable estimates of the capacity of the meadows to store and sequester carbon. I don't think this is the case – they allow you to budget the fate of seagrass C more accurately, but to determine the organic C stores or sequestration rates, you don't need the source information ?

Reply:We agree with the reviewer's comment and have rephrased this part in the ms.

29.L333: "ranged from": but then you provide an average +/- stdev.

Reply:Corrected.

30.L385 and elsewhere: when presenting a range of d13C values, provide the lowest (most negative value) first.

Reply:Corrected.

Please also note the supplement to this comment:
http://www.biogeosciences-discuss.net/bg-2016-131/bg-2016-131-AC1-supplement.pdf

———————————————————

[Figure]

**Supplement:**

[revised manuscript text omitted]

The article is distributed under the Creative Commons Attribution 3.0 License. Unless otherwise stated, associated published material is distributed under the same licence. Authors grant Copernicus Publications a license to publish the article and identify itself as the original publisher. Authors grant Copernicus Publications commercial rights to produce hardcopy

Field Code Changed volumes of the journal for purchase by libraries and individuals. Authors grant any third party the right to use the article freely under the stipulation that the original authors are given credit and the appropriate citation details are mentioned.

[revised manuscript text omitted]

---

## Referee Comment (RC2) · Anonymous Referee #2 · 14 Jul 2016

This study has collected a range of relevant data with which to provide estimates of organic carbon storage in Finland and Denmark. The data set would certainly be a useful addition to the growing literature of carbon storage in seagrass meadows and uniquely provides an indication of the range of carbon stocks observed for a single species of seagrass in different regions. The manuscript would require a major revision before being suitable for publication. The description of methods and presentation of results are confusing in terms of context values and units. Many are small points but collectively it becomes very frustrating to try and discern what has been done. The discussion needs to be kept focussed and relevant.

Specific comments

Study area: Line 126 the reference of Kristensen and Andersen 1987 seems at odds to

the preceeding sentences as it is a methodological paper that uses different mixtures of sand/eelgrass and shell to examine methods of analysis. A better reference would be Leipe T et al., 2011 Geo-Mar Lett 31, 175-188 who have analysed and obtained low PIC The inorganic content analysis on the samples could then be used to test that this is a common observation. In testing carbonate content, 10 samples were taken from each region, so was this from each station i.e. full coverage of sites? What are the units, wt%C or CaCO3? A line stating that that TC was therefore taken to represent OC is needed. Can the study sites be identified that made up the gradients of sheltered to exposed (see that on line 317 sorting is used to identify).

Field sampling: I cores taken to 40cm, why were C stocks only estimated to 25cm? Replicate cores taken at 5m for sediment but 15m apart for biomass? Clarify.

Seagrass and sediment variables: In determining the stable isotopic composition of above and below ground biomass and sediments it seems odd to use the term particulate organic carbon, just OC would be a better description. The method of d13C analysis is given but an estimation of sample analytical error for biomass and sediment is needed. It is unclear whether all OC data is derived from the mass spectrometric determination of d13C. Again error on sample analysis should be provided. It is confusing to know that the sediment has been analysed for OM and OC content and yet the OM content is not discussed only presented in Table 1. OC instead of OM should be presented in table 1. C stocks and accumulation: C stocks "obtained by depth integration of the POC mg C cm-3 of 0-25cm sediment layers would give units of mg C cm-2. which if multiplied by the areal extent of seagrass would give units of mg C, not g Corg m-2 as reported. Suddenly the notation has changed to Corg, need to be consistent. The areal extent of Danish seagrass is given but not the Finnish (30km2?). Economic loss would be better presented as loss in stock (pool) and rate of addition as separate values. Is the economic loss in the last 100 years is based on change areal extent times change in sequestration rate and biomass, but presumably no change in sediment C pool?

Results: If the C content of the biomass is known then why is the DW data used to determine the Corg pool in the living seagrass? It would be informative to have the dry bulk density data in Table 1 (rather than silt content) as it is not presented elsewhere. Given that OC data for the sediments is available, why are the OM results presented in the text? Especially as the OC stock section now presents the data as g C cm-3. Need to rationalise terms as it is confusing to use the same term Corg stocks (g C cm-3) in line 324 and (g C m-2) in line 329 and table 2. g C cm-3 (carbon density.The Corg accumulation section includes the sediment trap data. It is presented as gDW, not in OC and the relevance or significance of the measurements are not obvious. They would be better omitted. The areas represented by Finland, Limfjorden and Funen seagrass should be provided. In the isotope section the sediment and source data should be presented before the results of the mixing model. Allochthanous carbon is said to be a major contributor to the sediment (line 373), but no isotope value is given for its source term and it is not included as a component of Fig 5. It is not clear what it represents and is an input of terrestrial organic matter included or are there no significant riverine inputs to the area? Given the range of values for the sources it is difficult to understand how P. pectinatus and R. maritima can be isotopically distinguished from Z. marina in the mixing model for Finland

Discussion: Suggest discussion of sed trap data is removed as it adds little to the discussion. Analysis would be improved if it would highlight the differences in stock/accumulation, sources and particular influences (table 4)between Finland and Denmark given the difference in environmental factorsOverall the discussion could be shortened as there is a lot of extraneous text (more focussed) and consistent. One example is from the C accumulation and stock. Why not just compare with latest and most complete C stock assessment of Fourqurean et al., 2012 that has superceded all the other estimates. Are data for Z.marina included in the Fourqurean data set, if so, a comparison with this data would be more relevant. Table 3 C stock 627 g m-2, line 520 its 6900 t km-2, maths is wrong here? I do not see the relevance of comparing with tropical mangroves and marshes. In the comparison with stocks from Australia and

[Figure]

Asia this would only be appropriate if the stocks were all calculated in the top 25cms. Why are different units used for comparisons km2, m2 and ha-1. The discussion of cost of seagrass loss contains a lot of speculation and should be reduced. How viable is it to compare Danish stocks with those in Sweden and Germany? Line549 onwards Does this mean that it was assumed that all the carbon was lost from 25cm of sediment in the calculation? What was the basis for this assumption? Why is the loss being discussed now 50 years when it was 100 years in the results.

Further minor comments Line 290 291 state "proportion" but have reported %. When reporting mean values the error associated with the mean should also be reported. Line 291, 292 give mean %OC of biomass (that is used to determine C content ha-1) Line 293 change to C units. Line 306 The silt data is presented in Figure 2 and Table 1 (sorting data in Figure 2 but not referenced in the text). Line 326 In Figure 3 the max scale for Finland OC density is 20 and, but text reports value of 22 mg C cm-3, which must represent individual sediment values, it would be better if means for the site were reported in the text. Line 333 Finland stock ranged from – but only a single value given. Line 335 The min C stock for Denmark is different in the text to that given in Table 2. Line 356 should be annual areal Corg Line 373 allochthanous Line 383 were not where Line 473 does this data relate to drifting algal mats? Figure 1 Label a,b and c, which sites are shown? Figure 3 axis and legend is wrongly labelled as sediment profiles of particulate organic carbon content. This would have units of wt% or gC g-1. Figure 4 the legend "organic carbon stocks (Corg, g C m-2) "is confusing. Fig 5 It would be easier to discern the different source if patterns rather than shading were used. Table 2 column 5 should be Mt

---

## Author Response (AR1)

Support letter

Emilia Röhr
Åbo Akademi University
Tykistökatu 6
20520 Turku
Finland
mrohr@abo.fi

Associate Editor
Mr. Gerhard Herndl
Biogeoscienes
September 1st, 2016

Dear Editor

Please find attached the second thorough revision of our manuscript (bg-2016-131) entitled "Blue carbon stocks in Baltic Sea eelgrass (*Zostera marina*) meadows " by Maria Emilia Röhr, Christoffer Boström, Paula Canal- Vergés and Marianne Holmer to be considered for publication in Biogeosciences. Firstly, we would like to thank both of the reviewers as well as the Editor for their contribution and valuable input to the revision process of this manuscript. We found the comments most useful in further focusing and clarifying the manuscript text and tables.

We have now focused on re-analyzing our samples so, that in addition to the model that included both regions, we have ran a reduced, country wise DistLm analysis in which environmental drivers for Corg stocks were studied separately for Finland and Denmark. We have as well worked on harmonizing the terminology regarding the Blue Carbon calculations and results.
In addition, we have improved the style and data presented in tables 1 and 2 and figures and figure legends 1, 3 and 4.

We are including the detailed responses to the comments by Referee 2 in the section below together with revised manuscript with visible track changes. We have also attached a revised manuscript with no visible track changes to file manager. Our reply and answers to Referee 1 and the first version of the revised manuscript has been posted on the interactive discussion site earlier in summer 2016.

Looking forward to your response,

Thank you for your consideration!

Sincerely,

Emilia Röhr, MSc

Åbo Akademi University and University of Southern Denmark

Comments and answers to referee 2:

This study has collected a range of relevant data with which to provide estimates of organic carbon storage in Finland and Denmark. The data set would certainly be a useful addition to the growing literature of carbon storage in seagrass meadows and uniquely provides an indication of the range of carbon stocks observed for a single species of seagrass in different regions. The manuscript would require a major revision before being suitable for publication. The description of methods and presentation of results are confusing in terms of context values and units. Many are small points but collectively it becomes very frustrating to try and discern what has been done. The discussion needs to be kept focussed and relevant.

Specific comments:

1. Study area: Line 126 the reference of Kristensen and Andersen 1987 seems at odds to the proceeding sentences as it is a methodological paper that uses different mixtures of sand/eelgrass and shell to examine methods of analysis. A better reference would be Leipe T et al., 2011 Geo-Mar Lett 31, 175-188 who have analysed and obtained low PIC The inorganic content analysis on the samples could then be used to test that this is a common observation.

*Answer: Reference corrected (new line 125).*

2. In testing carbonate content, 10 samples were taken from each region, so was this from each station i.e. full coverage of sites?

*Answer: Yes, samples were from each of the 10 study sites in both regions. Info added in text line 126-127.*

3. What are the units, wt%C or CaCO3? A line stating that that TC was therefore taken to represent OC is needed.

*Answer: The value is given as percent based on the weight percent of IC and TC in the sample; e.g. the IC content was 0.01%DW and the TC content was 0.20%DW, accordingly the proportion of IC is 5% of TC (lines 125-129).*

4. Can the study sites be identified that made up the gradients of sheltered to exposed (see that on line 317 sorting is used to identify).

*Answer: For a comprehensive picture, sites were chosen to represent the full range of environmental conditions in each region (see Fig. 1). The sites were chosen on previous (>20 yrs) mapping projects of environmental settings of eelgrass beds in respective regions. To quantify the exposure regime in a standardized way, the organic content, degree of sorting and silt fraction gives an integrated measure of the type of setting eelgrass grows in (see Table 1, Fig. 2).*

Field sampling:

5. If cores taken to 40cm, why were C stocks only estimated to 25cm?

*Answer: We couldn't force the corer down to 40 cm depth at all of the sites therefor 0-25 cm was chosen to be the undisturbed depth section to be analyzed. In addition, 25 cm sediment cores have been widely used in similar studies and for normalization to the top 1 m of the sediments (eg. Fourqurean et al. 2012, Lavery et al. 2013). Comment added in lines 150-151.*

6. Replicate cores taken at 5m for sediment but 15m apart for biomass? Clarify.

*Answer: Corrected, 15 meters for both samplings, line 150.*

7. Seagrass and sediment variables: In determining the stable isotopic composition of above and below ground biomass and sediments it seems odd to use the term particulate organic carbon, just OC would be a better description.

*Answer: Corrected OC instead of POC for entire ms, tables and figures.*

8. The method of d13C analysis is given but an estimation of sample analytical error for biomass and sediment is needed. It is unclear whether all OC data is derived from the mass spectrometric determination of d13C. Again error on sample analysis should be provided. It is confusing to know that the sediment has been analyzed for OM and OC content and yet the OM content is not discussed only presented in Table 1. OC instead of OM should be presented in table 1.

*Answer: The analytical error (2.8 %) from the analysis has been added to the ms in line 369. The OC data is derived from the mass spectrometric determination of d13C. We have added a column including OC values to Table 1.*

9. C stocks and accumulation: line 220 →C stocks "obtained by depth integration of the POC mg C cm-3 of 0-25cm sediment layers would give units of mg C cm-2. which if multiplied by the areal extent of seagrass would give units of mg C, not g Corg m-2 as reported. Suddenly the notation has changed to Corg, need to be consistent.

*Answer: We have clarified this part in the ms line 210-214→ New text: "The $C_{org}$ (obtained by depth integration of the carbon density (mg C $cm^{-3}$) of 0-25 cm sediment layers (mg C $cm^{-2}$) of the sampled region was multiplied with estimated seagrass area of the region based on the most recent areal estimates (in $km^2$) of seagrass distribution available in the literature (Boström et al. 2014) and given as $C_{org}$ in g C $m^{-2}$."*

10. The areal extent of Danish seagrass is given but not the Finnish (30km2?).

*Answer: We have added a line stating the areal estimate for Finland (30 $km^2$) also to this section of the ms (line 214) the areal extent of Finnish meadows can also be seen in text line 330 and Table 2.*

11. Economic loss would be better presented as loss in stock (pool) and rate of addition as separate values.

*Answer: Unfortunately we do not understand this comment. Clarification is needed in order to potentially address something. In line 528 we have defined the magnitude of loss in $C_{org}$ stock.*

12. Is the economic loss in the last 100 years based on change areal extent times change in sequestration rate and biomass, but presumably no change in sediment C pool?

*Answer: Yes, we have measured the OC from existing eelgrass meadows and as there is no data available on the OC content of lost eelgrass meadows, we have assumed that if the lost meadows would have been present, the measured OC content would have been maintained.*

13. Results: If the C content of the biomass is known then why is the DW data used to determine the $C_{org}$ pool in the living seagrass?

*Answer: We have already corrected this part of the ms in connection with the Referee #1 comments. We use g DW OC m⁻² for determining the Corg pool, not the whole biomass (line:227)*

14. It would be informative to have the dry bulk density data in Table 1 (rather than silt content) as it is not presented elsewhere.

*Answer: Good point; sediment dry density has been added to Table 1, in addition to silt content.*

15. Given that OC data for the sediments is available, why are the OM results presented in the text? Especially as the OC stock section now presents the data as g C cm⁻³.

*Answer: The OM results are presented only as single sentence stating the average value of OM at each region (line 305 and 306). The average OC results for each region has been added to text in to lines 308-309 and OC values for each site have been added to Table 1.*

16. Need to rationalize terms as it is confusing to use the same term Corg stocks (g C cm-3) in line 324 and (g C m-2) in line 329 and Table 2. g C cm-3 (carbon density).

*Answer: Corg stock (mg C cm-3) has been re-named to "carbon density".*

17. The Corg accumulation section includes the sediment trap data. It is presented as gDW, not in OC and the relevance or significance of the measurements are not obvious. They would be better omitted.

*Answer: Sediment trap results have been omitted from the entire ms.*

18. The areas represented by Finland, Limfjorden and Funen seagrass should be provided.

*Answer: We have added the areal extent of Finnish meadows to line 214, the numbers are also given on l.330 (Finland) and 332(Denmark minimum and maximum) and in Table 2 (Finland, Funen, Limfjorden, Denmark minimum and maiximum areal extent).*

19. In the isotope section the sediment and source data should be presented before the results of the mixing model.

*Answer: The ms structure is now changed so that the sources are presented before the mixing model results.*

20. Allochthanous carbon is said to be a major contributor to the sediment (line 394-395), but no isotope value is given for its source term and it is not included as a component of Fig 5. It is not clear what it represents and is an input of terrestrial organic matter included or are there no significant riverine inputs to the area?

*Answer: "Allochtonous material" has been deleted from the ms. We have used phytoplankton values from plankton sieves or literature as source in Fig. 5 and text. Input of terrestrial organic matter was not included as a source. The Finnish sampling sites are in the archipelago, where riverine inputs are not significant sources of OM. Similarly, the study sites in Denmark are not in close proximity of rivers, therefor terrestrial input was not included as a source.*

21. Given the range of values for the sources it is difficult to understand how *P. pectinatus* and *R. maritima* can be isotopically distinguished from Z. marina in the mixing model for Finland

*Answer: R. maritima was not included in any models as it was not present at our study sites. R. cirrhosa contributed less than 5% to the sediment surface 13C signal at four Finnish sites and one Danish site and was included in the mixing model analysis when present. Lines 372-380 describe which species where present at the individual study sites. The values overlapping represent the range of the 13C signal in replicate samples for these species. When the 13C signals for each species were checked for individual sites, the signals for*

*different species differed significantly enough and running isotope mixing model analysis was thus possible to be made (eg.site Ängsö: Z. marina -10.12 to-10.50‰, R. cirrhosa -11.04‰, P. pectinatus -9.21‰).*

22. Discussion: Suggest discussion of sed trap data is removed as it adds little to the discussion.

*Answer: Sediment trap data has been removed.*

23. Analysis would be improved if it would highlight the differences in stock/accumulation, sources and particular influences (table 4) between Finland and Denmark given the difference in environmental factors

*Answer: We have re-analyzed our samples so, that in addition to the model that included both regions, we have ran a reduced, countrywise DistLm analysis in which environmental drivers for Corg stocks were studied separately for Finland and Denmark. We have added text explaining the results of these new analysis in to results section (lines 393-417) and to discussion section (lines 460-468).Our main finding in the new analysis was that biological variables (root to shoot- ratio, fraction of Z. marina from the sediment surface Corg pool) were important environmental drivers for Corg stocks in Denmark, while in Finland the sedimentary variables were better predictors for magnitude of Corg stocks.*

24. Overall the discussion could be shortened as there is a lot of extraneous text (more focussed) and consistent. One example is from the C accumulation and stock. Why not just compare with latest and most complete C stock assessment of Fourqurean et al., 2012 that has superceded all the other estimates.

*Answer: We have modified the discussion section so that it is more focused and consistent and the length has been reduced with 17 lines (10%).*

25. Are data for *Z. marina* included in the Fourqurean data set, if so, a comparison with this data would be more relevant.

*Answer: Fourqurean et al. 2012 does not report separate values for Z. marina.*

26. Table 3 C stock 627 g m-2, line 520its 6900 t km-2, maths is wrong here?

*Answer: The numbers have been corrected (line 521).*

27. I do not see the relevance of comparing with tropical mangroves and marshes.

*Answer: Comparison with mangroves and salt marsh has been removed from the ms.*

28. In the comparison with stocks from Australia and Asia this would only be appropriate if the stocks were all calculated in the top 25cms.

*Answer: Lavery et al. 2013 has indeed calculated the Corg stocks in the top 25 cm normalized to top 1 meter. Miyayima et al. 2015 has calculated the Corg stocks directly from 1 meter cores. Similarly to Lavery et al. 2013, we have calculated the Corg stocks we measured in the top 25 cm and extrapolated to 1m.*

29. Why are different units used for comparisons km2, m2 and ha-1.

*Answer: Different units have been used depending on the context, so that our numbers are comparable with numbers from other published papers that we are referring to.*

30. The discussion of cost of seagrass loss contains a lot of speculation and should be reduced. How viable is it to compare Danish stocks with those in Sweden and Germany? Line549 onwards

*Answer: We have focused and shortened this section in the ms and deleted comparison between our study regions and Swedish and German meadows (line 534-> of the old version of the ms).However, the comparison*

*was included in the first place because Danish meadows are generally very similar to those in Sweden and Germany while the Finnish meadows are perhaps outliers in this geographical context, and we found this interesting information to compare.*

31. Does this mean that it was assumed that all the carbon was lost from 25 cm of sediment in the calculation?

*Answer: Yes.*

32. What was the basis for this assumption?

*Answer: We have based this assumption on the fact that the erosion should be enhanced when the protection from the leaf-canopy and the rhizome-root mat is lost, and that the organic rich sediment should be easily transported away. Currently, we are aware of two studies (Serrano et al. 2016, Marba et al. 2014) regarding how much of the sediment in seagrass beds are eroded when beds are lost, both studies are conducted in the Australian Oyster harbor in Posidonia oceanica meadows. In addition, in the well-cited paper by Fourqurean et al. 2012 they assume that the top meter of the sediment is eroded and oxidized when a seagrass bed is lost, so in comparison with that study, 25 cm could be considered conservative estimate.*

33. Why is the loss being discussed now 50 years when it was 100 years in the results.

*Answer: We do not understand this comment as we find no such section in the discussion in which the losses would be discussed for 50 years.*

34. Further minor comments Line 290- 291 state "proportion" but have reported %.

*Answer: Has been corrected in to "fraction" (line 284).*

35. When reporting mean values the error associated with the mean should also be reported. Line 291, 292: give mean %OC of biomass (that is used to determine C content ha-1)

*Answer: We have added the mean % OC of biomass (lines 285 and 286). SEM values have been added to referred sentence (line 288-289).*

36. Line 293: change to C units.

*Answer: The units have been changed (line 231).*

37. Line 306: the silt data is presented in Figure 2 and Table 1 (sorting data in Figure 2 but not referenced in the text).

*Answer: Reference to fig. 2 has been added to line 315.*

38. Line 326: In Figure 3 the max scale for Finland OC density is 20 and, but text reports value of 22 mg C cm-3, which must represent individual sediment values, it would be better if means for the site were reported in the text.

*Answer: Values from individual replicates have been replaced with mean values (line 320 and 322).*

39. Line 333: Finland stock ranged from – but only a single value given.

*Answer: We think there is a misunderstanding here, since we found no such error in the ms. A single value is discussed and represented for Finland as there was only one eelgrass area estimate for Finland (line 330).*

40. Line 335: the min C stock for Denmark is different in the text to that given in Table 2.

*Answer: The values have been corrected to the ms lines 331-332.*

41. Line 356: should be annual areal Corg

*Answer: This part has been clarified in the text line 352.*

42. Line 373: allochthanous

*Answer: "Allochtanous" has been deleted from the ms.*

43. Line 383: were not where

*Answer: "Where" has been corrected to "were" (line 367).*

44. Line 473: does this data relate to drifting algal mats?

*Answer: This sentence (currently lines 485-488) is not actually relating to drifting algal mats, but rather describing the importance of understanding that the sediment OM pool is almost always a mix of different sources and it is very unlikely that the OM pool would consist only of a sole source of OM.*

45. Figure 1: label a, b and c, which sites are shown?

*Answer: Labelling has been added to the figure.*

46. Figure 3: axis and legend is wrongly labelled as sediment profiles of particulate organic carbon content. This would have units of wt% or gC g-1.

*Answer: The axis and legend has been changed in to C mg C cm-3 in Fig. 3.*

47. Figure 4: the legend "organic carbon stocks (Corg, g C m-2) "is confusing.

*Answer: The legend for Fig. 4 has been changed to Corg stocks (g C m$^{-2}$).*

48. Fig 5: It would be easier to discern the different source if patterns rather than shading were used.

*Answer: We find shading is visually easier to interpret, since many of the sections are small and using patterns for these sections makes the figure more "messy" and harder to interpret. Since the fig. legend is organized exactly in the same order as the bars, it makes it very difficult to mix the different sources.*

49. Table 2: column 5 should be Mt.

*Answer: This has been corrected to the table 2.*

[revised manuscript text omitted]

The article is distributed under the Creative Commons Attribution 3.0 License. Unless otherwise stated, associated published material is distributed under the same licence. Authors grant Copernicus Publications a license to publish the article and identify itself as the original publisher. Authors grant Copernicus Publications commercial rights to produce hardcopy volumes of the journal for purchase by libraries and individuals. Authors grant any third party the right to use the article freely under the stipulation that the original authors are given credit and the appropriate citation details are mentioned.

[revised manuscript text omitted]

---

## Author Response (AR2)

Röhr et al. rebuttal letter Oct 9 2016

Dear Editor, We have now addressed the minor comments by the referee and hope now that the ms can be published. Looking forward to your respons,

Sincerely,

Emilia Röhr

**Suggestions for revision or reasons for rejection (will be published if the paper is accepted for final publication)**
The authors have addressed many of the issues raised during the first reviews, but a number of issues remain problematic :

1. The fact that samples were not acidified prior to analysis remains not well discussed. In the original version, the authors stated that "The inorganic carbon content in our samples was low (0.003 to 0.3 %DW, n= 10 per region) and therefore carbonates were not removed from the sediment samples prior to the analysis". Given that %OC varies from 0.1 to 5.78 % (average values) for the different sites, this implies that inorganic carbon can contribute at least 5.2 % of the total carbon (assuming the highest %IC was found in the sample with the highest %OC), but it could be more if %IC and %OC are not related. Hence, I feel the data should be presented so that readers can make their own judgement on how problematic this is. Assuming the carbonate fraction has a d13C of around 0 per mil and the organic fraction somewhere between -15 and -20 per mil, a 5% contribution of carbonate to in the measured sample can correspond to a bias of about 1 per mil, which is large. This should at least be acknowledged and discussed.

*Answer: We understand the referees concern. The %IC varied between 0.3-5.6% (average 3.3%) and was on average less than 0.76‰ (range 0.16-1.17‰) which is less or similar to the natural variance of the sediment isotopic signal. As we have addressed in our previous reply, it is not trivial to acidify the samples, in particular in low organic content samples (such as those sampled from Finland) where acidification itself can cause similar analytical errors than the procedure would be able to eliminate (Schlacher and Connolly 2014). To further clarify this issue, we have now added a sentence to the ms (l. 129-132) in which we have acknowledged this possible bias.*

2. Line 334 and further in the revised version: this still does not make sense. The total organic C stock is measured and should be derived from your bulk density and %OC data, the OC accumulation rates are irrelevant here. The same applies to the section on line 223 and further in the revised version (see also my original comments). The same goes for their reply to my original comment (12) that you use a rate to calculate the OC stock.

*Answer:  The Corg stocks are calculated according to Lavery et al. (2013) by multiplying the OC (OC mg/gDW) measured from different sections of the sediment core with the sediment dry bulk density (g/cm³), the product of this calculation gives units in OC mgC/cm³. The product is further multiplied with the depth of the measured sediment section, and all the products from the*

*different sections of the sediment core were summed (depth integrated Corg stocks). The product of this summing was further multiplied with 10 in order to get the units into C g DW/ $m^2$. We have now added a section to the ms where this procedure is explained in more detail (l. 213-216).*

*From line 342 → our goal is to estimate how much carbon has been lost from the Danish eelgrass meadows due to reduced eelgrass area in the last 100 years. For that calculation OC accumulation rate is actually relevant. In order to address the lost carbon stocks, we need to have a number for both annual sediment accumulation rate (we have used 3 different rates, minimum, medium and maximum, obtained from literature) as well as an estimate of rate on how long it has taken to accumulate the average Corg stock of the area in the top 25 cm of sediment (eg. depth integrated Corg stocks in top 25 cm in Finland: 627 C gDW/$m^2$) with the sedimentation rate of 2.02 mm/y (the medium rate), the product of this calculation is referred to as annual areal carbon accumulation (t C $ha^{-1}$ $y^{-1}$) and the procedure is explained in detail in the ms in lines 231-237.*

3. My original comment concerning the relationship observed between %OC and bulk density; the authors mention they don't understand this comment and mention their data show exactly this. Yes, they do, but the point is that this is not a causal relationship but a logical inverse relationship reflecting different contributions of organic matter (low density) and the mineral fraction (which has a high density). Thus, statements that %OC is determined partially by density are not appropriate.

*Answer: We do understand the referees concern on this subject, however, we do not find such a section in our manuscript where our text would imply that % OC would be partially determined by sediment density, but rather that according to the DistLm model sediment dry density was an important predictor variable for variation in the Corg stocks. We understand that high organic content sediments naturally have lower densities and thus, the results from the DistLm model can be logically explained by the inverse relationship of sediment dry density and the mineral fraction.*

4. Earlier comment regarding extremely low OC contents (<0.001 mg cm-3), I still do not see these low numbers on Figure 3. These are incredibly low.

*Answer: As stated in our previous reply, the number <0.001 stands for those sites that had replicates in which the OC was below detection limit of the elemental analyzer (%OC < 0.001% DW). Therefore, no number could be plotted in the figure for those replicates and hence, cannot be seen in Fig. 3. We have added a sentence to the figure legend stating that numbers below detection limit are not shown in the Figure. The values plotted in the Figure are averages of C (mg C cm-3) from each site presented with standard error of mean.*

Minor comments:

5. Table 1: use "." as the decimal separator throughout, not comma's.
*Answer: Corrected.*

6. Table 1: use 1 decimal only to report d13C data given the analytical uncertainty.
*Answer: Corrected.*